# AUTODISCOVERY:
# Open-ended Scientific Discovery via Bayesian Surprise

**Dhruv Agarwal**$^{*\alpha}$ **Bodhisattwa Prasad Majumder**$^{*\beta}$

**Reece Adamson**$^{*\alpha}$ **Megha Chakravorty**$^{*\alpha}$ **Satvika Reddy Gavireddy**$^{*\alpha}$
**Aditya Parashar**$^{\gamma}$ **Harshit Surana**$^{\beta}$ **Bhavana Dalvi Mishra**$^{\beta}$

**Andrew McCallum**$^{\alpha}$ **Ashish Sabharwal**$^{\beta}$ **Peter Clark**$^{\beta}$

$^{\alpha}$University of Massachusetts Amherst  $^{\beta}$Allen Institute for AI  $^{\gamma}$Capital One
`dagarwal@cs.umass.edu, bodhisattwam@allenai.org`
$^{*}$equal contributions

֍ `https://github.com/allenai/autodiscovery`

## Abstract

The promise of autonomous scientific discovery (ASD) hinges not only on answering questions, but also on knowing which questions to ask. Most recent works in ASD explore the use of large language models (LLMs) in goal-driven settings, relying on human-specified research questions to guide hypothesis generation. However, scientific discovery may be accelerated further by allowing the AI system to drive exploration by its own criteria. The few existing approaches in open-ended ASD select hypotheses based on diversity heuristics or subjective proxies for human interestingness, but the former struggles to meaningfully navigate the typically vast hypothesis space, and the latter suffers from imprecise definitions. This paper presents AUTODISCOVERY—a method for open-ended ASD that instead drives scientific exploration using *Bayesian surprise*. Here, we quantify the epistemic shift from the LLM's prior beliefs about a hypothesis to its posterior beliefs after gathering experimental results. To efficiently explore the space of nested hypotheses, our method employs a Monte Carlo tree search (MCTS) strategy with progressive widening using surprisal as the reward function. We evaluate AUTODISCOVERY in the setting of data-driven discovery across 21 real-world datasets spanning domains such as biology, economics, finance, and behavioral science. Our results demonstrate that under a fixed budget, AUTODISCOVERY substantially outperforms competitors by producing 5-29% more discoveries deemed surprising by the LLM. Our human evaluation further reveals that two-thirds of discoveries made by our system are surprising to domain experts as well, suggesting this is an important step towards building open-ended ASD systems.

## 1 Introduction

There has been a surge of recent progress in using large language models (LLMs) for autonomous scientific discovery (ASD) [Majumder et al., 2024b, Wang et al., 2024, Lu et al., 2024, Skarlinski et al., 2024, Majumder et al., 2025, Gottweis et al., 2025, Huang et al., 2025]. Most prior works operate within a *"goal-driven"* setting: given some data[1], the user is required to provide a research

---

[1]A collection of datasets (data-driven discovery) or related papers (literature-driven discovery).

39th Conference on Neural Information Processing Systems (NeurIPS 2025).

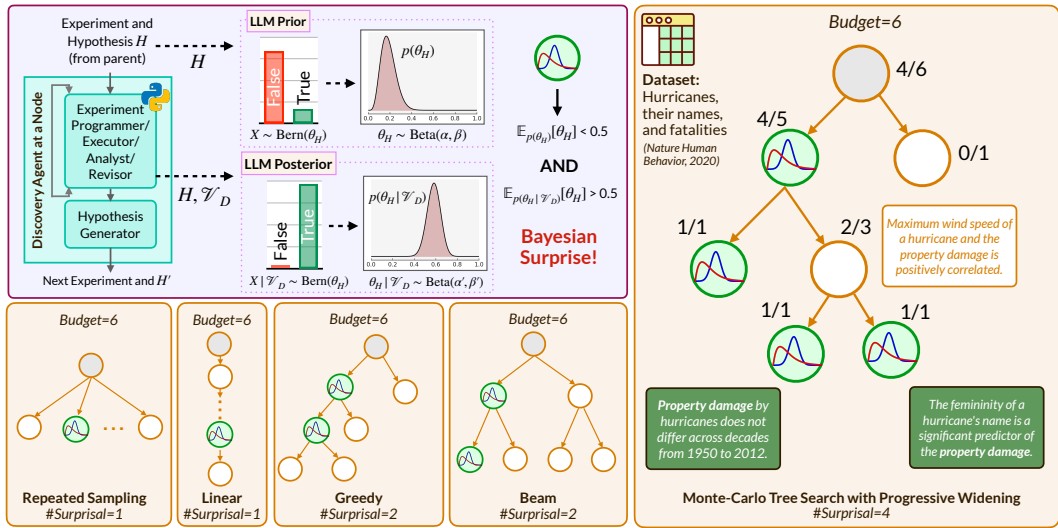

Figure 1: **Overview of AUTODISCOVERY:** A method for open-ended ASD that is guided by *Bayesian surprise*. We elicit LLM prior and posterior beliefs about hypotheses via sampling, and use surprisal as a reward function within an MCTS procedure to find hypotheses by trading-off exploration and exploitation of the hypothesis space in search for surprising discoveries.

question; then an LLM is prompted to (1) generate a hypothesis (i.e., an assertion about the true state of the world) that is relevant to the research question, (2) propose an experiment to test the hypothesis, (3) generate and execute code to perform the experiment, and (4) analyze the experiment results to derive a conclusion.

On the other hand, while there has been some work on ideating research ideas from the literature [Spangler et al., 2014, Baek et al., 2025], there has been limited investigation of the full *"open-ended"* setting, where the ASD system itself explores more broadly by generating hypotheses according to its own measures of research promise, executing the aforementioned steps, and then using its results to propose new hypotheses in a never-ending process (akin to the workflow of a human scientist). A key challenge, then, is that of search—*which hypotheses should be investigated next that will lead to novel, impactful scientific discoveries?*

Prior efforts in open-ended automated discovery have used search strategies such as rejection sampling and evolutionary algorithms with either a human in the loop [Yamada et al., 2025, Gottweis et al., 2025, Jansen et al., 2025] to filter the generated hypotheses (thus, not fully autonomous) or using automatic rewards such as diversity and LLM-as-judge proxies for human interestingness, novelty, or utility [Zhang et al., 2023a, Lu et al., 2024]. Diversity alone, however, is insufficient due to the massive search space of hypotheses in real-world scientific domains, where not all regions may equally be likely to lead to discoveries. Moreover, the ability to sample diverse sequences from an LLM has itself been shown to be challenging [Lanchantin et al., 2025, Krishnamurthy et al., 2024]. Human proxy metrics, e.g., interestingness, are not suitable either due to their subjective nature, with even human scientists demonstrating a high degree of disagreement [Ceci and Peters, 1982, Cicchetti, 1991, Rothwell and Martyn, 2000, Weller, 2001, Pier et al., 2018], making their automated proxies unreliable. Therefore, it remains unclear: *how can diverse hypotheses be explored at scale and what automatic metrics may guide scientific discovery?*

In this work, we address these questions and propose **AUTODISCOVERY**—a method for open-ended ASD that is guided by *Bayesian surprise* [Itti and Baldi, 2005], which quantifies how data affects a natural or artificial observer by measuring the distance between its posterior and prior belief distributions (see Fig. 1). Our choice is motivated by recent findings from Shi and Evans [2023], which show that the improbability or surprisal of a hypothesis is often a strong predictor of scientific impact. To automate the computation of surprisal, we use an LLM model itself as the Bayesian observer. In so doing, we make a simplifying assumption and focus on developing a procedure to expand the knowledge frontier of the model itself.[2] We mechanize this frontier by deriving prior and

---

[2]We anticipate that this knowledge frontier will rapidly approach that of humans as models, especially retrieval-augmented ones, continue to advance.

posterior distributions about an LLM's belief about hypotheses, without and with conditioning on an empirical evaluation of the hypotheses given data, respectively.

To sample hypotheses with high surprisal, we propose a Monte-Carlo tree search (MCTS) [Coulom, 2006] procedure with progressive widening [Couëtoux et al., 2011], which provides a principled mechanism to balance exploration and exploitation of the vast hypothesis search space. In Figure 1, we show how MCTS is able to navigate the search space and find the most number of surprising hypotheses under a fixed budget, where surprisals may even be discovered from non-surprising nodes.

We evaluate AUTODISCOVERY in data-driven discovery (DDD) [Majumder et al., 2024b, 2025], where the input to a discovery task is a collection of datasets and a programming environment,[3] making it suitable for evaluating fully autonomous discovery. In experiments over 21 real-world datasets across behavioral science, economics, biology, and finance, we find that AUTODISCOVERY finds 5-29% more number of hypotheses surprising to the LLM agent as compared to strong search baselines. In a human study, we further find that two-thirds of surprising discoveries found by AUTODISCOVERY correlate with human surprisal, indicating that optimizing for Bayesian surprise may be an effective proxy to guide real-world open-ended ASD.

In summary, our contributions are as follows:

- We provide the first formal definition of surprise within the context of autonomous scientific discovery, inspired by prior work on Bayesian surprise.
- We present a novel method, AutoDS, that combines this notion of surprise with MCTS to perform hypothesis search in an open-ended setting (no goal specified).
- We show that AUTODISCOVERY outperforms competitors by 5-29% at finding discoveries that are surprising to the LLM in extensive data-driven discovery experiments, spanning 21 real-world datasets. In a human study with 500+ hypotheses, we find that 67% of the surprising discoveries made by AUTODISCOVERY are surprising to domain experts, showing promise for real-world open-ended ASD.

## 2    Preliminaries

**DDD formalization.** Following Majumder et al. [2025], we define a *data-driven hypothesis $H$* in $\mathcal{H}$ (the space of such hypotheses) as a natural language statement that defines relationships ($r$) between a set of variables ($v$) under contexts ($c$). Further, given a dataset $D$, the truth value for $H$ may be inferred using a verification procedure $\mathcal{V}_D : \mathcal{H} \rightarrow \{\text{supported}, \text{unsupported}\}$, where the space of valid $\mathcal{V}_D$ is potentially any executable Python program.

**Discovery agent.** We call any agent capable of generating and verifying a data-driven hypothesis given a dataset and a programming environment a discovery agent. In this work, we use a multi-agent architecture [Majumder et al., 2024a], composed of LLMs that collaboratively propose experiment plans, write and execute Python code to conduct those experiments, critique and fix mistakes, and analyze their results. Figure 1 shows the input/output flow for our discovery agent. Please see the appendix for the complete details.

**Open-ended DDD.** Unlike in the goal-driven setting, where the task is to search for a verifiable hypothesis that may answer a research question provided explicitly as input, the open-ended setting requires a discovery agent to iteratively generate and verify hypotheses given only the dataset to make discoveries. To improve search efficiency, an exploration strategy may be used that repeatedly invokes the discovery agent in its inner loop and typically terminates after a predefined budget is exhausted.

## 3    AUTODISCOVERY: Autonomous Discovery via Surprisal

We present AUTODISCOVERY (Fig. 1), a method for open-ended autonomous scientific discovery that leverages LLMs to identify and prioritize hypotheses based on surprisal—a principled measure of belief shift induced by experimental evidence. To this end, we formalize surprisal using a Bayesian framework, introduce a practical method for belief elicitation via LLM sampling, and describe how surprisal can guide efficient exploration of the hypothesis space via tree-based search.

---

[3]No wet lab experiments.

### 3.1 Measuring Surprisal

Consider a dataset $D$, a data-driven hypothesis $H \in \mathcal{H}$, and its verification procedure $\mathcal{V}_D$. For a given agent, let $\theta_H \in [0, 1]$ denote its *belief* about the support for $H$ in $D$, i.e., a probability that $H$ may be verifiable by some $\mathcal{V}_D$. We assume the agent is uncertain about the value of $\theta_H$ and we model this uncertainly using a Beta distribution, i.e., $\theta_H \sim \text{Beta}(\alpha, \beta)$. In particular, let $P(\theta_H)$ denote the agent's prior (Beta) distribution for the value of $\theta_H$ given only the hypothesis, and $P(\theta_H \mid \mathcal{V}_D)$ its posterior (Beta) distribution after observing results from the verification procedure $\mathcal{V}_D$.

We first discuss how to estimate these two distributions by querying the agent, and then describe how to use these estimated distributions to compute the agent's surprisal.

**Belief Elicitation via Sampling.** In order to empirically estimate an LLM agent's prior and posterior distributions of $\theta_H$, we use the Beta-Bernoulli conjugacy to propose a simple procedure that samples $n$ boolean responses from the LLM about the truth value for $H$, before and after revealing results from $\mathcal{V}_D(H)$ in the prompt. The empirical frequencies of *"true"* responses ($k_{\text{prior}}$ and $k_{\text{post}}$) are treated as Bernoulli outcomes, which are used to make Bayesian updates[4] to estimate the prior and posterior distributions of $\theta_H$, denoted $P_{\text{est}}(\theta_H)$ and $P_{\text{est}}(\theta_H \mid \mathcal{V}_D)$, respectively, as follows:

$$P_{\text{est}}(\theta_H) := \text{Beta}(\theta_H \mid 1 + k_{\text{prior}}, 1 + n - k_{\text{prior}}), \text{ and} \tag{1}$$

$$P_{\text{est}}(\theta_H \mid \mathcal{V}_D) := \text{Beta}(\theta_H \mid 1 + k_{\text{prior}} + k_{\text{post}}, 1 + (n - k_{\text{prior}}) + (n - k_{\text{post}})). \tag{2}$$

We will henceforth use $P_{\text{est}}(\theta_H)$ and $P_{\text{est}}(\theta_H \mid \mathcal{V}_D)$ as empirical estimates of $P(\theta_H)$ and $P(\theta_H \mid \mathcal{V}_D)$.

**Bayesian surprise.** Inspired by Itti and Baldi [2005], we propose the use of Bayesian surprise, a distance measure between the prior and posterior beliefs, to quantify the magnitude of change in beliefs that occurs when a discovery agent observes results from $\mathcal{V}_D(H)$. Specifically, we define

$$\text{BS}(H, \mathcal{V}_D) := D_{\text{KL}}(P(\theta_H \mid \mathcal{V}_D) \parallel P(\theta_H)). \tag{3}$$

**Surprisal.** We now formalize the intuition that surprisal arises when beliefs update in a directionally significant way. We say that a surprisal has occurred if there is a *shift* in the agent's expected belief about $H$, i.e., a change in leaning about its beliefs (e.g., from supported to unsupported), given evidence from $\mathcal{V}_D$; specifically if $\mathbb{E}_{P(\theta_H \mid \mathcal{V}_D)}[\theta_H]$ lies on a different side of a decision threshold $\delta$ (typically, we set $\delta = 0.5$) than $\mathbb{E}_{P(\theta_H)}[\theta_H]$. To capture both the surprise due to directional change and its informational significance, we define *Bayesian surprise under belief shift*[5] as

$$\text{BS}_{\text{shift}}(H, \mathcal{V}_D) := \begin{cases} \text{BS}(H, \mathcal{V}_D), & \text{if } (\mathbb{E}_{P(\theta_H \mid \mathcal{V}_D)}[\theta_H] - \delta)(\mathbb{E}_{P(\theta_H)}[\theta_H] - \delta) \leq 0 \\ & \quad \wedge \ \mathbb{E}_{P(\theta_H \mid \mathcal{V}_D)}[\theta_H] \neq \mathbb{E}_{P(\theta_H)}[\theta_H] \\ 0, & \text{otherwise.} \end{cases} \tag{4}$$

We can then formally define *surprisal* as an indicator function

$$\text{S}(H, \mathcal{V}_D) := \mathbb{1}[\text{BS}_{\text{shift}}(H, \mathcal{V}_D) > 0], \tag{5}$$

which captures whether a belief shift about a hypothesis has occurred on observing new evidence.

### 3.2 Search using Surprisal-driven MCTS

Our goal is to expand the knowledge frontier of the LLM by proposing hypotheses that yield surprisal under verification. However, naïve strategies, such as repeated independent sampling and greedy search by an LLM, (a) struggle to reliably generate diverse hypotheses, and (b) do not optimally balance exploration and exploitation of the vast hypothesis search space $\mathcal{H}$.

To address both problems, AUTODISCOVERY uses **Monte Carlo tree search (MCTS)** [Coulom, 2006] guided by **surprisal** as the reward function. In particular, we build a hierarchy of diverse hypotheses by iteratively conditioning the LLM on a branch of hypothesis sequences composed of prior discoveries to sample $k$ new experiments to investigate further. To prioritize the expansion of nodes that may be more likely to yield surprisal, we use upper-confidence bound on trees (UCT) [Kocsis and Szepesvári, 2006] as a principled method to trade off exploration and exploitation, a strategy commonly applied in large, combinatorial spaces (e.g., game playing [Gelly et al., 2012] and program synthesis [Lim and Yoo, 2016]).

---

[4]We assume an uninformed prior $\text{Beta}(1, 1)$ when no hypothesis is provided.

[5]The first conditional clause ensures that the expected prior and posterior beliefs lie on different sides of $\delta$, while the second ensures that they are not both equal to $\delta$.

**Algorithm 1** MCTS with Progressive Widening

---

**Require:** $k \in \mathbb{R}_{>0}; \alpha \in [0, 1]$
1: **procedure** EXPAND($H_{\text{parent}}$)
2:     **if** $|\text{children}(H_{\text{parent}})| < kN(H_{\text{parent}})^{\alpha}$ **then**
        ▷ Progressive Widening
3:         $H \sim \text{LLM}(\cdot \mid \{h \in \text{path}(H_{\text{parent}} \rightsquigarrow \text{root})\})$
4:         $\text{children}(H_{\text{parent}}).\text{add}(H)$
5:         **return** $H$
6:     **else**
7:         $H \leftarrow \arg\max_{h \in \text{children}(H_{\text{parent}})} \text{UCT}(h)$
8:         **return** EXPAND($H$)
9:     **end if**
10: **end procedure**

---

**Procedure.** We build a search tree where each node represents a hypothesis $H \in \mathcal{H}$, and edges correspond to sampling steps executed by a discovery agent to generate new hypotheses. The algorithm proceeds in four phases in each iteration:

1. **Selection:** Starting from the root, the tree is traversed by selecting a node $H_{\text{parent}}$ for expansion that represents a region with high potential for surprisal. In particular, we use the UCT acquisition function as described in Eq. (6), where $N(H)$ is the number of visits to any node in the subtree rooted at $H$, i.e, $\text{subtree}(H)$, and $C$ is a tunable constant that controls the strength of exploratory behavior. The first term computes the average surprisal from node $H$ and encourages exploitation of known good regions, while the second term encourages exploration of new nodes.

2. **Expansion:** We then sample a child hypothesis $H$ by conditioning on all prior experiments and results in the branch from $H_{\text{parent}}$ to the root (see Algorithm 1) Since it is intractable to sample all possible children at a node, we employ **progressive widening** [Couëtoux et al., 2011], which dynamically increases the number of children a node must have based on its visitation count. Importantly, this encourages search to revisit multiple promising regions within the search space before expanding any one of them in an unbalanced manner.

3. **Execution[6]:** The sampled hypothesis $H$ from expansion is evaluated by executing its corresponding $\mathcal{V}_D$ and estimating its surprisal $\text{S}(H, \mathcal{V}_D)$ using the belief elicitation procedure (§ 3.1).

4. **Backpropagation:** The estimated surprisal is propagated back through the tree from $H$ to the root, updating surprisal and visitation statistics for each node in the path.

$$\text{UCT}(H) = \underbrace{\frac{\sum_{h \in \text{subtree}(H)} \text{S}(h, \mathcal{V}_D^{(h)})}{N(H)}}_{\text{Exploit}} + \underbrace{C \cdot \sqrt{\frac{2 \log N(H_{\text{parent}})}{N(H)}}}_{\text{Explore}} \tag{6}$$

## 3.3 Deduplication via LLM-based HAC

Despite incorporating a search strategy to guide discovery, hypothesis generation in AUTODISCOVERY may sample semantic duplicates. To identify these, we propose an LLM-based hierarchical agglomerative clustering (HAC) procedure (inspired by Zhang et al. [2023b]) that combines similarity within a textual embedding space with LLM reasoning to identify semantically equivalent hypotheses. We run this procedure once after the search budget is exhausted.

We start by constructing an HAC tree using text embeddings of hypotheses. For every merge decision between a pair of clusters identified in the HAC linkage matrix, two representative hypotheses—each with its structured breakdown of context, variables, and relationships—are passed to an LLM (GPT-4o, in our experiments) to determine whether they are semantically equivalent. Specifically, we sample a boolean response from the LLM about whether the structured hypotheses are equivalent. If the proportion of *"true"* responses exceeds 0.7, we merge the clusters and propagate the updated assignment before proceeding with the next linkage step. If it does not, the clusters remain independent. The iteration proceeds until no further merges remain to be evaluated, either because all candidate pairs involve clusters whose descendants have already been labeled non-duplicates or because the LLM has reviewed every remaining cluster pair.

---

[6]Note that our departure from the "simulation" step is motivated by the fact that execution of $\mathcal{V}_D(H)$ (a) is inexpensive and (b) does not alter the state for subsequent actions in our setting (unlike, e.g., in game playing).

# 4 Experiments

Our empirical evaluation assesses the effectiveness of various methods for the task of open-ended DDD. The input for the task is a dataset $D$, its associated metadata, and a budget (which we set to 500) specifying the total number of hypotheses the agent is allowed to explore and verify. The goal of the agent is to discover as many surprising, but verifiable (over $D$), hypotheses as possible. We assess performance on this based on (a) the number of unique hypotheses generated, and (b) the number of surprisals they produce under the fixed experiment budget.

## 4.1 Datasets

We utilize a total of 21 datasets ($D$) from the following benchmark sources spanning areas such as biology, economics, finance, and behavioral science. We selected the range of datasets to maximize data-shape heterogeneity, scientific salience (associated with top-tier publications), and breadth of domains in our evaluation.

- **DiscoveryBench** [Majumder et al., 2025], a comprehensive benchmark designed to assess the ability of large language models to autonomously search for and verify hypotheses using associated datasets. DiscoveryBench comprises 264 real-world discovery tasks sourced from published papers across six domains (e.g., sociology, engineering) spanning across 14 scientific datasets. We selected the following five datasets and associated metadata from DiscoveryBench as a representative sample: `freshwater-fish`, `nls-bmi`, `nls-ses`, `nls-incarceration`, and `requirement-engineering`.
- **BLADE** [Gu et al., 2024] is a benchmark evaluating language agents on justifiable scientific data analysis using real-world datasets and expert-defined analysis decisions. We use all 15 datasets from BLADE in our work: `affairs`, `amtl`, `boxes`, `caschools`, `conversation`, `crofoot`, `fertility`, `fish`, `hurricane`, `mortgage`, `panda_nuts`, `reading`, `soccer`, `teachingratings`, and `toy`.
- **SEA-AD** [Hawrylycz et al., 2024] is a multimodal cellular atlas of the human brain across the Alzheimer's disease spectrum, developed by the Allen Institute. We utilize the donor-level metadata, including demographic details, cognitive status, and neuropathological assessments.

## 4.2 Baselines

We rigorously evaluate our method against common repeated sampling baselines as well as tree-based search methods. To ensure a fair comparison, we keep the following constant across methods—the reward function (here, surprisal) and the exploration budget (500 in all experiments). All baselines and AUTODISCOVERY use the same discovery agent with `GPT-4o`. The discovery agent has access to a Python environment with available statistical and data analysis packages (e.g., `sklearn`), and can generate Python code to run in an execution environment to verify a hypothesis. The agent can also install additional Python packages to successfully execute a proposed verification experiment.

- **Repeated (independent) sampling** generates hypotheses in a parallel, context-free manner, i.e. without knowledge of other experimental results within the same run, using ancestral sampling. (schematic in "Repeated Sampling"; Figure 1).
- **Last-$K$ (linear) sampling** is a context-aware version of repeated sampling with a strictly sequential exploration strategy in which each new experiment directly follows from the most recent one, forming a single, linear path of reasoning. To accommodate the context length limitation of LLMs, we retain only the last $K = 100$ parent nodes as context during hypothesis generation. See search schematic in "Linear"; Figure 1.
- **Greedy tree search** is one of two tree-based search baselines we evaluate. It focuses on exploitation by always selecting the highest-value node at each step to condition on for hypothesis generation, resulting in a narrow, semi-linear search path (schematic in "Greedy"; Figure 1). This translates to an MCTS variant with the exploration constant $C = 0$.
- **Beam search** is a tree-based exploration strategy (inspired by [Li et al., 2025]) that restricts search at each level by retaining only the top-$b$ candidate states (i.e., beam width $b$), ranked by surprisal and visitation statistics. We set both the branching factor and beam width to 8. At every level of the tree, 64 candidate states are generated, and the top 8 are retained for expansion at the next level. Unlike MCTS, beam search performs breadth-first expansion and aggresively prunes the search tree, making it more sensitive to early ranking errors (schematic in "Beam"; Figure 1).

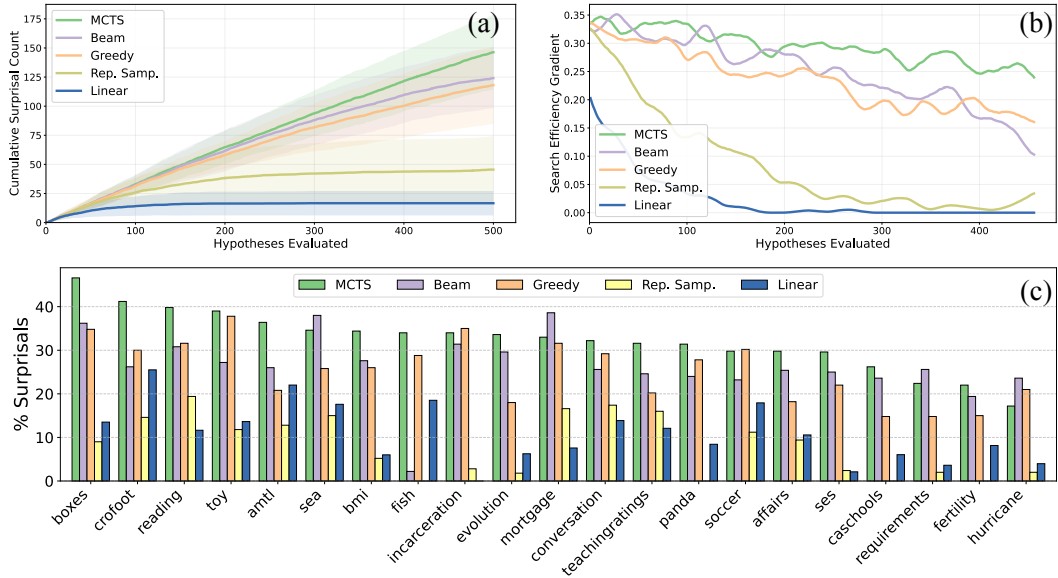

Figure 2: **Search Performance. (a)** Cumulative number of surprisals discovered across timesteps within a budget of 500 evaluations, averaged over 21 datasets. **(b)** Search efficiency gradient computed using a sliding window of 10 iterations. **(c)** Number of surprisals discovered per dataset. **Takeaway:** AUTODISCOVERY outperforms baselines, including other tree-search methods, in both search efficiency and number of surprisals discovered.

## 5 Results and Discussion

We first demonstrate via human studies the correlation between LLM and human surprisal and showcase also a correlation with human interestingness and utility. We then show a comparison between our proposed Bayesian surprise and alternative automatic proxies as reward functions to guide search using MCTS. We then compare AUTODISCOVERY with strong search baselines on the ability to optimize for surprisal. Finally, we manually validate the accuracy of various aspects within the discovery agent workflow. Complete experimental details are provided in the appendix.

### 5.1 Bayesian Surprise versus other Automatic Rewards

We run an analysis of AUTODISCOVERY swapping out the Bayesian surprise reward for other automatic metrics that are commonly used to assess hypothesis quality in ASD—LLM interestingness and LLM utility. We also include direct LLM surprisal to provide a comparison with an alternative mechanism for surprisal elicitation. Each of these metrics is operationalized by prompting the LLM to provide $n$ boolean responses to their respective questions, e.g., *"is this hypothesis interesting to you?"*, then using the average number of *"yes"* responses as a reward in MCTS.

**Bayesian surprise correlates with expert surprisal.** We assess performance by evaluating how often the hypotheses generated using each automatic reward result in *human* surprisal. Each hypothesis is annotated by 3 STEM MS/PhDs with a total of 1,620 LLM surprisal hypotheses pooled from 4 datasets across MCTS runs with four automatic rewards. Our results in Table 2 show that across annotator familiarity, AUTODISCOVERY with Bayesian surprise finds hypotheses that have a much higher correlation (67%) with human surprisal than any of the other automatic rewards, with a 95% confidence interval of [0.63, 0.71] computed using bootstrapping with 10,000 re-samples.

| Reward | H. Interesting | H. Utility |
|---|---|---|
| Bayesian Surprise | 0.73 | 0.79 |
| LLM Surprisal | **0.76** | **0.80** |
| LLM Interestingness | 0.74 | 0.78 |
| LLM Utility | 0.73 | 0.78 |

Table 1: **Human Interestingness and Utility Scores Across Automatic Rewards.** Average human ratings for interestingness and utility for four automatic rewards used in MCTS.

| Familiarity | Bayesian Surprise | LLM Surprisal | LLM Interesting | LLM Utility | IAA |
|---|---|---|---|---|---|
| Low | **0.64** | 0.13 | 0.18 | 0.22 | 0.61 |
| Medium | **0.66** | 0.12 | 0.13 | 0.24 | 0.51 |
| High | **0.62** | 0.08 | 0.14 | 0.19 | 0.71 |
| Overall | **0.67** | 0.11 | 0.15 | 0.21 | - |

Table 2: **Human Surprisal Across Automatic Rewards.** Average human surprisal ratings across different annotator familiarity levels for four automatic rewards used in MCTS: Bayesian surprise, direct surprisal, interesting(ness), and utility with inter-annotator agreement (IAA). **Takeaway:** Bayesian surprise results in the highest number of human surprisals, with the next best reward showing an average score lower by 0.46 points.

**Interestingness and utility lack clear semantics.** We ran another study to assess whether these automatic metrics correlate with other human notions such as interestingness and utility. Our results in Table 1 show that though Bayesian surprise clearly does, so do all the other metrics. This suggests that eliciting human interestingness and utility may be difficult due to their subjective nature. A corollary of this is that deriving automatic versions of such metrics without clear semantics is likely not useful for guiding open-ended ASD.

## 5.2 Optimizing for Bayesian Surprisal

**MCTS outperforms other search strategies.** In Fig. 2(a), we show the cumulative number of surprisals discovered across timesteps, averaged over all datasets. We find that all tree search baselines outperform repeated sampling and linear search, with MCTS in AUTODISCOVERY showing the highest efficiency for discovery as well as the highest number of surprisals collected.

Notably, as shown in Fig. 2(b), MCTS shows minimal reduction in search efficiency across time, unlike other baselines, including greedy tree search and beam search. This indicates room for scaling AUTODISCOVERY with a higher budget to continue collecting surprising discoveries. In Fig. 2(c), we show that the aggregate trend in search performance holds across the evaluated datasets, with AUTODISCOVERY (MCTS) showing the best performance in 17 out of 21 datasets.

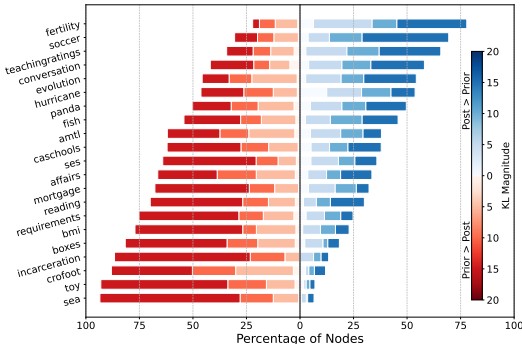

**LLM beliefs shift differently across domains.** In Fig. 3, we plot $BS_{shift}(\cdot, \cdot)$, i.e. the KL divergence between the prior and posterior beliefs of surprisals, for the hypotheses found using AUTODISCOVERY along with the direction of the belief shift. Our analysis reveals different directional tendencies across domains, with a

Figure 3: **Belief shift across datasets.** Bayesian surprise under belief shift for surprisals discovered using AUTODISCOVERY, grouped by domain and direction of shift.

higher proportion of surprisals shifting from supported to unsupported. We also observe lower KL divergence when the model's beliefs shift towards supporting the hypothesis, possibly indicating the need for greater evidence in confirmatory studies than in falsification ones [Huang et al., 2025].

## 5.3 Validating the Discovery Agent Framework

To evaluate the faithfulness of the verification in AUTODISCOVERY, we evaluate two critical pieces: (1) whether the experiment is valid, i.e., whether it can be implemented with the available data and helps to confirm the hypothesis, and (2) whether the experiment was faithfully implemented. We sampled 175 nodes for the MCTS tree run on `nls_bmi` from DiscoveryBench with associ-

| Artifact | % Validity | IAA |
|---|---|---|
| Experiment | 98.58 | 0.97 |
| Implementation | 98.01 | 0.98 |
| Deduplication | 90.76 | 0.75 |

Table 3: Discovery agent validity

ated experiments, code, and analysis, and asked two annotators to annotate the artifacts for experiment

and implementation validity. Table 3 demonstrates very high validity (>95%) for both experiment and implementation artifacts, also with a very high inter-annotator agreement (Gwet's AC1 as >0.95).

To evaluate the deduplication via LLM-based HAC pipeline, we subsampled 120 pairs of hypotheses from MCTS trees on all five DiscoveryBench datasets. Two hypotheses in a pair are sampled from the same HAC clusters. We asked a range of 1-3 annotators if the two hypotheses from a pair are structurally equivalent or not. Table 3 shows clusters found by our LLM-based HAC method indeed group duplicate hypotheses with 91% validity, with good annotator agreement (Gwet's AC1 as 0.75).

# 6  Additional Related Work

As noted earlier, there has been an explosion of interest in AI-assisted/automated discovery in the last few years, e.g., AIScientist [Yamada et al., 2025], CodeScientist [Jansen et al., 2025], AgentLab [Schmidgall et al., 2025], Popper [Huang et al., 2025], HypoBench [Liu et al., 2025]. However, these systems are mainly goal-driven, performing start-to-finish experimentation given a clear research goal, rather than iterative, open-ended, goal-free exploration (our context). While there has been some work on initial hypothesis generation, in particular from the literature [Baek et al., 2025, Spangler et al., 2014], our goal is iterative generation and search over a large space.

While our framework is general, we have evaluated it in the context of data-driven discovery, a rich context for science [Majumder et al., 2024b, 2025, Gu et al., 2024]. While some commercial tools, e.g., [Bailis et al., 2017], offer programmatic ways to exhaustively sweep a size-bounded hypothesis space, our goal is different, namely, to heuristically search a much larger space using LLM.

More generally, surprise (or equivalently, failed expectations) have played a historically important role in AI, leading to work on encoding expectations and failure-driven learning [Riesbeck, 1981, Schank and Abelson, 1988, Schank, 1983] (indeed, almost all learning can be viewed as responding to failed expectations). We adopt a formal notion of surprise here and show how it can be successfully applied for guiding open-ended exploration. We also note the close connection between Bayesian surprise and other metrics from information theory, e.g., mutual information [Gao et al., 2024].

# 7  Conclusion

We introduce a formal framework for Bayesian surprise in autonomous scientific discovery and propose AUTODISCOVERY, a method for open-ended scientific discovery that uses this framework alongside MCTS to find hypotheses that can expand an LLM's knowledge frontier. Through evaluations across 21 real-world datasets and a comprehensive human study, we demonstrate that AUTODISCOVERY not only outperforms strong baselines in making surprising discoveries but also aligns well with human judgements of surprise. While we remain cautious about open-ended AI systems for scientific discovery without sufficient guardrails, academic skepticism, and peer review, our results underscore the potential benefit a system such as AUTODISCOVERY may provide in accelerating science.

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

# Appendix

## 8 Discovery Agents

### 8.1 Finite State Machine

When evaluating a hypothesis, agents work collaboratively within a shared context. Speaker selection is determined by a finite state machine where transitions are determined based on the prior speaker and the content of the last messsage. Algorithm 2 provides pseudocode for the speaker selection transitions.

---
**Algorithm 2** Speaker Selection Algorithm

---
```
 1: procedure SELECT_NEXT_SPEAKER(last_speaker, last_response)
 2:     if last_speaker = "user_proxy" then
 3:         return hypothesis_generator
 4:     else if last_speaker = "hypothesis_generator" then
 5:         return experiment_programmer
 6:     else if last_speaker = "experiment_programmer" then
 7:         return code_executor
 8:     else if last_speaker = "code_executor" then
 9:         return experiment_analyst
10:     else if last_speaker = "experiment_analyst" then
11:         if last_response.error = True and code_failure_count < 6 then
12:             code_failure_count ← self.code_failure_count + 1
13:             return experiment_programmer
14:         else
15:             self.code_failure_count ← 0
16:             return experiment_reviewer
17:         end if
18:     else if last_speaker = "experiment_reviewer" then
19:         if last_response.error = True and self.experiment_revision_count < 1 then
20:             self.experiment_revision_count ← self.experiment_revision_count + 1
21:             return experiment_reviser
22:         else
23:             self.experiment_revision_count ← 0
24:             return experiment_generator
25:         end if
26:     end if
27:     if last_speaker = "experiment_reviser" then
28:         return experiment_programmer
29:     else if last_speaker = "experiment_generator" then
30:         return None
31:     end if
32: end procedure
```

---

### 8.2 LLM Agents

AUTODISCOVERY defines the following agents:

- **Experiment Generator:** Produces experiment plans for evaluating a hypothesis

- **Hypothesis Generator:** Proposes a hypothesis which predicts the outcome of an experiment.

- **Experiment Programmer**: Writes code to implement an experiment plan. The programmer is allowed 6 attempts to correctly implement the experiment based on feedback from the Experiment Analyst.

- **Code Executor**: Executes code produced by the Experiment Programmer; returns the exit code and Standard Output. (Not LLM-based).

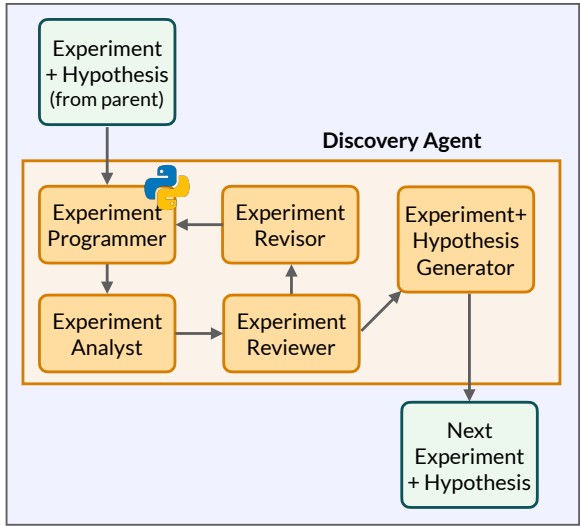

Figure 4: Finite state machine for the discovery agent.

- **Experiment Analyst**: Analyzes code execution output. Provides feedback to Experiment Programmer if code needs adjustments.

- **Experiment Reviewer:** Reviews experiment results for alignment with original experiment plan and ability to support validation of the hypothesis.

- **Experiment Reviser:** Produces a revised experiment should the experiment or its implementation fail to be successfully implemented or allow validation of the hypothesis.

- **Hypothesis Belief:** Evaluates whether the hypothesis is believed true based on only the hypothesis statement (prior) or the hypothesis statement and associated experimental results (posterior). A temperature of 0.7 was used for the experiments underlying the results presented.

- **Image Analyst:** Provides textual description of images produced by code from Experiment Programmer.

### 8.2.1 System Prompts

The system prompts for each agent are listed below:

**Experiment Generator**

You are a curious researcher who is interested in open-ended research based on the provided dataset. Think of a creative and interesting new experiment/analysis to conduct. Do not provide the code yourself but explain in natural language what the experiment should do for a programmer.Remember, you are interested in open-ended research, so do not hesitate to design experiments/analyses that do not directly relate to the previous one.
Here are a few instructions that you must follow: 1. Strictly use only the dataset provided and do not create synthetic data or columns that cannot be derived from the given columns. 2. Each experiment should be creative, independent, and self contained.3. Check prior experiments, especially look through any recommendations made to improve the richness of the hypothesis and consider this information while proposing new experiments. However, do not repeat the same experiment plan.Here are a few suggestions that will help you to create creative new experiments: 1. You are encouraged to create composite attributes derived from the given columns. This is formally known as feature engineering in the machine learning literature. 2. You are encouraged to propose experiments involving complex statistical tests. Remember, your programmer can install arbitrary python packages which would allow it write code for complex statistical analysis. For example: propose appropriate tests involving categorical variables. 3. You are encouraged to propose experiments involving complex predictive or causal models. For example: propose non linear predictive models such as gradient boost trees or SVM as appropriate to model multivariate relationships. 4. You are encouraged to propose experiments that only focus on a subset of the given dataset. This will help create unique interesting context to validate a hypothesis. For example: if dataset has multiple categorical variable, you could split the data based on specific values of such variable which would then allow you to validate a hypothesis in that specific context.
Generally, In a typical data-driven discovery workflow, you may need to explore and visualize the data for possible high-level insights, clean, transform, or derive new variables from the dataset to be suited for the investigation, deep dive into specific parts of the data for fine-grained analysis, perform data modeling and statistical tests.

**Experiment Programmer**

You will generate code based on an experiment description. State is not preserved between code blocks. Your code will be included in a python file that is executed. You must explicitly print any relevant results to standard out appropriately. Anything that you want displayed must be printed to standard out or presented using plt.showMake sure you return code in the proper format to execute, i.e. python code.Ensure your code is clean and concise, and include debug statements only when they are absolutely necessary.Use only the dataset given and do not assume any other files are available. Import any libraries you need to use. Always attempt to import a library first in case it is already installed. You may install libraries if they are not already available. If you need to install a library, use the following code example:

```python
import subprocess
import sys

def install(package):
    subprocess.check_call([sys.executable, "-m", "pip", "install", "--quiet",
    package])
```

When installing python packages use the --quiet option to minimize unnecessary outputPrefer using installed libraries over installing new libraries whenever possible. If possible, instead of downgrading library versions, try to adapt your code to work with a more updated version that is already installed. Never attempt to create a new environment. Always use the current environment. If the code requires generating plots, use plt.show (not plt.savefig). Avoid printing the whole data structure to the console directly if it is large; instead, print concise results which directly address the experimentYou are allowed 6 total attempts to run the code, including debugging attempts. Debugging Instructions: 1. Only debug if you are either unsure about the executability of the code or the validity of the code satisfying the proposed experiment.2. If the code you are writing is intended for debugging purposes, you MUST clearly tag it using a comment line that contains only "[debug]".3. DO NOT use "[debug]" anywhere in your code when you are sure about your implementation. 4. DO NOT combine debug code and actual implementation of the experiment, keep them seperate.5. For each experiment, you are allowed to debug each 3 times.6. It is still good to minimise the number of debugging steps.

## Experiment Analyst

You are responsible for analyzing the execution output generated by the programmer. If no code was executed, indicate that there was an error.If the code includes a line # [debug] i.e "[debug]" as a comment, strictly treat this as a failed or debugging experiment. In such cases strictly return error status as **true**, provide information that it was a debug code execution, take feedback and request the experiment to be retried with the new information. Otherwise, you should analyze the results and provide a short summary of the findings from the current experiment.

## Experiment Reviewer

You are responsible to holitically review the generated code, the output, and the analysis w.r.t the original experiment plan.Assess whether the experiment was faithfully implemented. The implementation follows the experiment plan without any significant deviations.A successful experiment should have clear results that can be interpreted irrespective of the fact that it supports or rejects the initial hypothesis.If there were issues, provide feedback on what went wrong.

## Hypothesis Generator

Propose a hypothesis which predicts the outcome of the experiment. The hypothesis should be a statement that can be tested by the experiment. Provide the context, variables, and relationships that are relevant to the hypothesis. The context should be a set of boundary conditions for the hypothesis. The variables should be the concepts that interact in a meaningful way under the context to produce the hypothesis. The relationships should be the interactions between the variables under the context that produces the hypothesis.Keep relationships concise. e.g., "inversely proportional", "positive quadratic", "significant predictor", "causally mediating", to name a few.

## Experiment Reviser

You are a curious researcher revisiting the most recent hypothesis that could not be validated effectively in the previous experiment which eventually failed as indicated by the experiment_reviewer. Your goal is to revise thismost recent failed experiment by addressing the weaknesses/limitations given by the experiment_reviewer. The revised experiment should still aim to validate the most recent hypothesis. Do not provide the code yourself but explain in natural language what the experiment should do for a programmer.

Here are a few instructions that you must follow: 1. Strictly use only the dataset
provided and do not create synthetic data or columns that cannot be derived from
the given columns. 2. You must consider the most recent failed experiment and the
feedback and revise accordingly so that it is effective to validate the most recent
hypothesis. Here are a few generic suggestions that will help you to revise the
experiment along with the feedback you received from experiment_reviewer: 1. You
are encouraged to revise experiments to include focused analysis on a subset of the
dataset, using feature engineering techniques where appropriate. 2. You are
encouraged to revise experiments to retain complex statistical analyses, leveraging
external Python packages if needed to support sophisticated methods.For example:
revise appropriate tests involving categorical variables. 3. You are encouraged to
revise experiments invovling complex predictive or causal models. For example:
revise non linear predictive models such as gradient boost trees or SVM as
appropriate to model multivariate relationships. 4. You are encouraged to revise
experiments that only focus on a subset of the given dataset. This will help create
unique interesting context to validate a hypothesis. For example: if dataset has
multiple categorical variable, you could split the data based on specific values of
such variable which would then allow you to validate a hypothesis in that specific
context.
Generally, In a typical data-driven discovery workflow, you may need to explore and
visualize the data for possible high-level insights, clean, transform, or derive new
variables from the dataset to be suited for the investigation, deep dive into
specific parts of the data for fine-grained analysis, perform data modeling and
statistical tests.

## Hypothesis Belief

You are a belief distribution agent that evaluates the latest hypothesis. Based on
the available evidence from prior experiments, assess whether the hypothesis is
true or false. Respond with a JSON object: {"believes_hypothesis": true} or
{"believes_hypothesis": false}.
Hypothesis: {hypothesis}
Carefully consider the evidence and reasoning before making your assessment. Be
critical in your evaluation, but fair to the evidence presented.

## Image Analyst

Analyze the given plot image and provide the following:

1. Plot Type: Identify the type of plot (e.g., heatmap, bar plot, scatter plot) and
its purpose.
2. Axes:
    * Titles and labels, including units.
    * Value ranges for both axes.
3. Data Trends:
    * For scatter plots: note trends, clusters, or outliers.
    * For bar plots: highlight the tallest and shortest bars and patterns.
    * For heatmaps: identify areas of high and low values.
4. Statistical Insights: Mention any relevant statistics if applicable.
5. Annotations and Legends: Describe key annotations or legends.
6. Overall Impression: Summarize insights and implications for further analysis.
7. Interpretation: Provide insights or perspectives based on the data presented.
What conclusions can be drawn?
8. User Objective: If applicable, address the user's question or objective related
to the image.
9. Limitations: Discuss any limitations or biases in the data that could affect
conclusions.

## 8.3 Deduplication (Clustering)

**Prompt**

```
You are given two sets of hypotheses. Each set describes a context, the variables
involved, and the statistical relationships between them. Your task is to determine
if both sets indicate the same statistical behavior. Consider the following:
Context: The conditions or boundaries under which the relationship holds. Both sets
must have identical contexts.
Variables: All variables must match. Even if their names differ, they must refer to
the same concept.
Relationships: Each hypothesis may include one or more pairs of explanatory and
response variables. The statistical relationship between these variables must be
equivalent, regardless of how it is described.
Your answer should be either "Yes" or "No" with no additional explanation.
Hypothesis Set 1:
{hypothesis_1}
Hypothesis Set 2:
{hypothesis_2}
Answer:
```

## 8.4 Agent Hyperparameters

| Parameter | Value/Setting | |
|---|---|---|
| Model | $\begin{cases} \text{Image Analyst} \\ \text{otherwise} \end{cases}$ | $\begin{matrix} \text{GPT-4o} \\ \{\text{GPT-4o, o4-mini}\} \end{matrix}$ |
| Temperature | $\begin{cases} \text{Image Analyst} \\ \text{Hypothesis Belief} \\ \text{o4-mini} \\ \text{otherwise} \end{cases}$ | $\begin{matrix} 1.0 \\ 0.7 \\ \text{NA} \\ 0 \end{matrix}$ |
| Timeout | 600 seconds | |
| Max Network Retries | 3 | |
| Response Caching | Disabled | |
| Number of Belief Samples | $\begin{cases} \text{GPT-4o} \\ \text{o4-mini} \end{cases}$ | $\begin{matrix} 30 \\ 8 \end{matrix}$ |
| Maximum Context Tokens | 128,000 | |
| Maximum Message Tokens | 4,096 | |
| Number Revisal Attempts | 1 | |
| Number of Code Attempts | 6 | |

Table 4: System Configuration Parameters

# 9 Baselines: Search Algorithms

## 9.1 Repeated Sampling

Repeated sampling is achieved by deriving all experiments independently from the root of the tree, i.e., all nodes have only a single ancestor node. Repeated sampling can be seen as a special case of MCTS in AUTODISCOVERY, by either disabling progressive widening, or using progressive widening with sufficiently large constants, e.g. $k \geq$ sampling budget.

## 9.2 Linear Search

Linear search conditions subsequent experiments on prior hypotheses in a single experimental trajectory. MCTS can be configured to enable this type of search by setting appropriate constants, e.g., $k = 0.5, \alpha = 0$, which constrains each node to have no more than a single child.

### 9.3 Greedy Tree Search

Greedy tree search focuses on exploitation by always selecting the highest-value node at each step to condition on for hypothesis generation, resulting in a narrow, semi-linear search path. This translates to an MCTS variant with the exploration constant $C = 0$.

### 9.4 Beam Search

The experimental results presented utilize a beam width $\beta = 8$ and branching factor $k = 8$.

---

**Algorithm 3** Beam Search

---

**Require:** beam width $\beta$, branching factor $k$
    **procedure** SAMPLE($s$)
2:      **for all** $s \in$ beam **do**
          exps $\leftarrow$ untried[$s$]
4:          **for all** $a \in$ exps **do**
              $s' \sim T(s, a)$
6:             children[$s$].add($s'$)
          **end for**
8:          children[$s$] $\leftarrow$ sort$\big($children[$s$]$\big)$ by $\dfrac{\text{w}}{\text{n}}$
          beam $\leftarrow$ children[$s$][1:$\beta$]
10:    **end for**
    **end procedure**

---

### 9.5 Programmatic Search: an alternative to LLM agents?

A bottleneck of LLM-driven hypothesis search and verification is latency due to API calls used for both hypothesis and code generation/debugging for verification. Average time per node/hypothesis is 75 seconds, with a maximum of 600 seconds in some of the AUTODISCOVERY trees. However, particularly for data-driven discovery, Bailis et al. [2017] developed a heuristic-based programmatic system with efficient sampling that can ingest up to 2M data events per second. Inspired by this and to consider an alternative to AUTODISCOVERY in the setting of DDD, we developed a deterministic programmatic search baseline devoid of LLM calls. The system exhaustively enumerates up to a million contexts (with enough data coverage) and performs pre-written (often shallow, e.g., correlation analysis) statistical analyses in under ten minutes.

To compute Bayesian surprise over these programmatically generated insights, we use an LLM to translate them into a hypothesis statement. Despite being mathematically unique, many insights are semantically similar, especially hypotheses with the same interacting variables and relationship, computed across exploded contexts, but essentially encode the same generalized insight. After an LLM-based deduplication, the programmatic search generated an average of 109 unique surprisals, when computed post-hoc. While limited due to dataset-specific human interventions and shallow insights, programmatic search can empower the initial explorations done by AUTODISCOVERY with 10× speed while LLM-driven hypothesis generation (and verification) can focus on complex hypotheses beyond the scope of the programmatic search—we leave this as a future work.

**Algorithm 4** Programmatic Search Baseline

---

**Require:** Dataset $\mathcal{D}$, target $y$, categorical set $\mathcal{C}$, numeric set $\mathcal{N}$, max depth $d_{\max}$, coverage threshold $\tau$, significance level $\alpha$

**Ensure:** Ranked insight list $\mathcal{I}$

1: $\mathcal{C}_{\text{bin}} \leftarrow \text{QUANTILEBIN}(\mathcal{N})$
   $\triangleright$ bin numerics
2: $\mathcal{F} \leftarrow \mathcal{C} \cup \mathcal{C}_{\text{bin}}$
3: $\mathcal{M} \leftarrow \{\}$
   $\triangleright$ valid context masks
4: **for** $k \leftarrow 1$ **to** $d_{\max}$ **do**
5:     **for all** feature subsets $s \subseteq \mathcal{F}$ with $|s| = k$ **do**
6:         **for all** level vectors $\ell$ of $s$ **do**
7:             $m \leftarrow \bigwedge_{(f,v)\in(s,\ell)}(f = v)$
8:             **if** $\text{RELATIVEFREQ}(m) \geq \tau$ **then**
9:                 $\mathcal{M} \leftarrow \mathcal{M} \cup \{m\}$
10:             **end if**
11:         **end for**
12:     **end for**
13: **end for**
14: $\mathcal{M} \leftarrow \mathcal{M} \cup \{\text{full-data mask}\}$

       $\triangleright$ — First-pass statistics —
15: $\mathcal{R}_1 \leftarrow \{\}$
16: **for all** $m \in \mathcal{M}$ **do**
17:     $\mathcal{D}_m \leftarrow \mathcal{D}[m]$
18:     $\mathcal{R}_1 \leftarrow \mathcal{R}_1 \cup \text{SINGLEFACTOR}(\mathcal{D}_m, y)$
19:     $\mathcal{R}_1 \leftarrow \mathcal{R}_1 \cup \text{CORRELATION}(\mathcal{D}_m, y)$
20:     $\mathcal{R}_1 \leftarrow \mathcal{R}_1 \cup \text{SIGNIFICANTDRIVERS}(\mathcal{D}_m, y)$
21: **end for**
22: $\mathcal{R}_1 \leftarrow \text{FILTERSORT}(\mathcal{R}_1, \alpha, \tau)$

       $\triangleright$ — Second-pass pattern mining —
23: $\mathcal{R}_2 \leftarrow \text{DETECTFLIPSEMERGENCE}(\mathcal{R}_1)$
24: $\mathcal{R}_2 \leftarrow \text{FILTERSORT}(\mathcal{R}_2, \alpha, \tau)$

       $\triangleright$ — Symbolic rendering —
25: $\mathcal{I} \leftarrow \text{RENDERMATHSTRINGS}(\mathcal{R}_1 \cup \mathcal{R}_2)$ **return** $\mathcal{I}$

---

# 10 Limitations

- **Pitfalls with agentic frameworks.** Providing LLM agents agency over experimental direction and implementation can lead to unexpected failures, e.g., corruption of the execution environment by installing improper libraries.
- **Context window corruption/limits.** Particularly in our setting of open-ended discovery, the discovery agent may show aberrant behavior if the context either becomes exceedingly long or corrupted due to, e.g., unexpected code execution output. Nodes with many ancestors (e.g. >150) can also suffer from context-overflowing and show a similar failure mode.
- **Ungrounded generations.** LLMs occasionally introduce attributes not grounded in the dataset, with prompt-based constraints insufficient to address the issue completely. Without human oversight, this may result in the generation of scientifically spurious findings, which is important to guard against.
- **LLM surprisal vs. human surprisal.** Despite the 67% agreement between Bayesian surprise and our human evaluation (Table 2), surprising the LLM is only a proxy for novelty to humans. To reduce this gap, future work may consider literature-grounded extensions that explicitly incorporate information such as citations into the LLM surprisal computation.

# 11 Human Annotations

## 11.1 Qualtrics/Prolific Annotation Flow

We conducted our human annotation study using the Qualtrics platform with annotator recruitment and access managed via Prolific (Fig. 5). Each participant was assigned to a specific dataset through a unique Prolific link. For every dataset, we created a distinct Qualtrics survey set, comprising 20 hypotheses, and three independent annotators were assigned to evaluate each set.

The annotation flow within Qualtrics was structured as follows:

1. **Prolific ID Capture:** At the beginning of the survey, participants were asked to enter their unique Prolific ID to ensure traceability and consistency of responses.

2. **Dataset Exposure and Familiarity Assessment:** Before any hypothesis was shown, participants were introduced to the dataset context and structure. They were then asked to indicate their familiarity level with the dataset and its domain using the following prompt:

   *How familiar are you with the dataset and its domain?*
   - Not familiar at all
   - Slightly familiar
   - Moderately familiar
   - Very familiar
   - Extremely familiar

   Following this, participants rated their **confidence in their familiarity score** on a slider ranging from 0 (no confidence) to 100 (complete confidence):

   *What is your confidence in the familiarity score you provided above?* (Slider: 0–100)

3. **Hypothesis Evaluation:** Participants were then presented with a sequence of 20 hypotheses, shown one at a time. For each hypothesis, they responded to the following four questions using sliders ranging from 0 to 1. Here, 0 indicates *False*, 1 indicates *True*, and 0.5 represents *Unsure*:

   (a) *What is your belief about this hypothesis?* (Slider: 0 = False, 0.5 = Unsure, 1 = True)
   (b) *Could knowing the truth value for this hypothesis be useful to you?* (Slider: 0 = False, 0.5 = Unsure, 1 = True)
   (c) *Could knowing the truth value for this hypothesis be useful to the scientific community?* (Slider: 0 = False, 0.5 = Unsure, 1 = True)
   (d) *Do you think this hypothesis is interesting?* (Slider: 0 = False, 0.5 = Unsure, 1 = True)

   All slider responses were recorded as real-valued scores in the range $[0, 1]$ for subsequent quantitative analysis.

To manage participant recruitment and validate response completeness, we used the Prolific platform. Each annotator accessed a specific Qualtrics survey set through a unique Prolific link tied to the hypothesis sets.

At the end of the survey, participants were shown a **completion code**, which they were instructed to submit on Prolific to confirm they had fully completed the task. This code allowed us to:

- Verify that only participants who completed the entire annotation process were compensated,
- Match survey responses with Prolific IDs for traceability,
- Filter out any incomplete or prematurely exited surveys to ensure data quality.

Only responses associated with a valid completion code were included in our final analysis.

**Annotator population and payment**   Evaluating scientific hypotheses is a knowledge-intensive task. To obtain high-quality, meaningful annotations on the hypotheses, we screen participants and only allow those with a Master's/PhD degree in Mathematics and statistics, Information and Communication Technologies, Engineering, manufacturing and construction, or Natural sciences, to

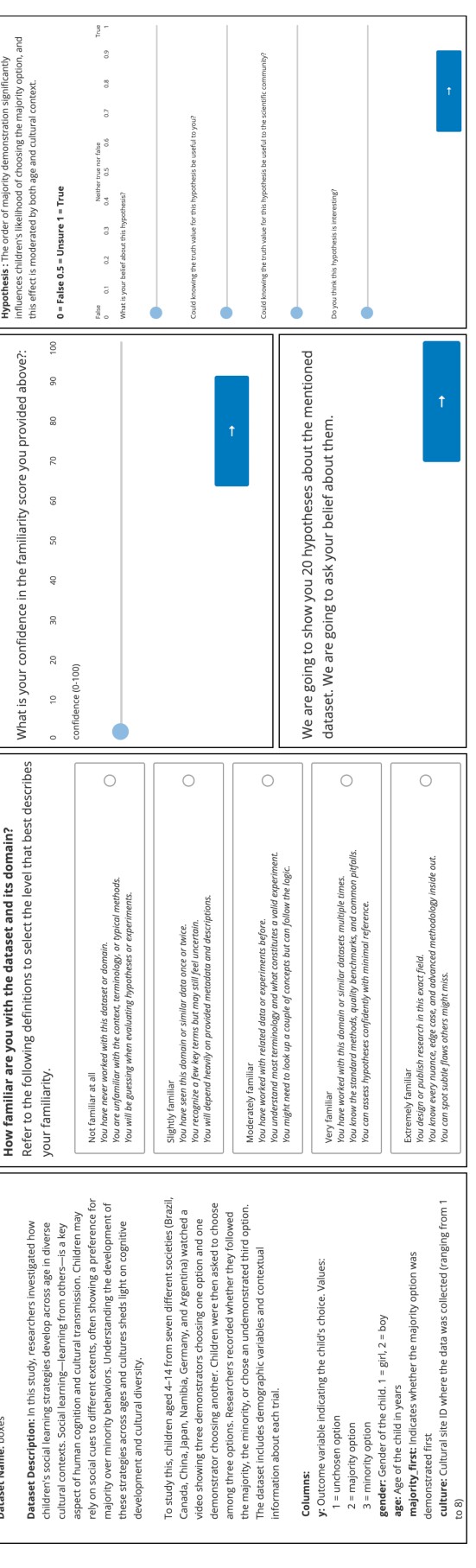

Figure 5: Annotation interface with Qualtrics/Prolific.

take part in the evaluation task. Other pre-screen qualifications were: Residence to be US, UK, or Canada, and they should be fluent in English. We set an hourly pay rate of $16.68, and we anticipate taking on an average of 20 minutes (which turned out to be the median completion time, when computed post-experiment) to complete the survey. For each unique Qualtrics link (each having 20 hypotheses), we require each participant to be unique too, however, the same participant can take part in two or more unique Qualtrics links.

## 11.2 Internal Annotation

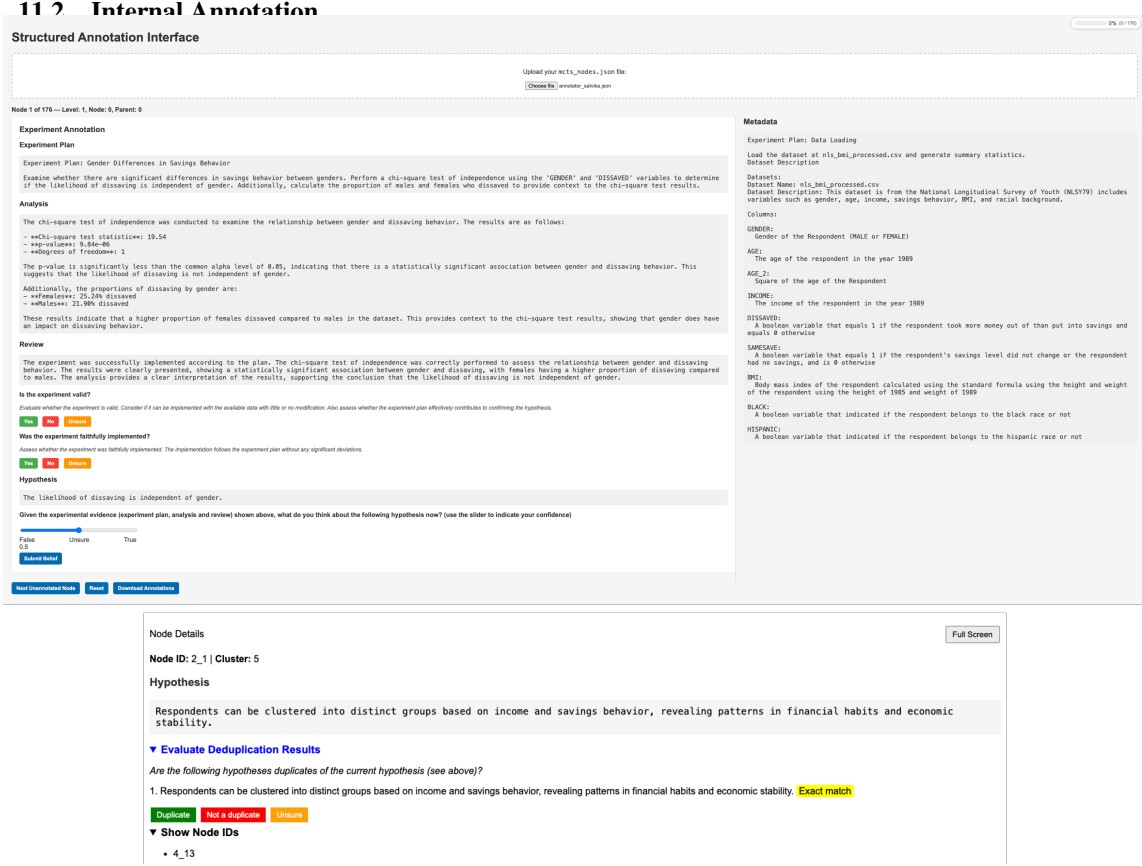

Figure 6: Internal annotation tool for discovery agent and deduplication verification.

To verify the faithfulness of the different components within our discovery pipeline, we additionally built an internal annotation tool. Specifically, we verify three key pieces: (1) whether the **experiment** proposed by the LLM is **valid**, i.e. its feasibility to be implemented using the available data and its ability to confirm the hypothesis, (2) whether the Python code **implementation** is **faithful** to the proposed experiment, and (3) whether the **deduplication** procedure is valid. Each author annotator was shown the experiment plan, experiment analysis, and a review summary for hypotheses (Fig. 6). Based on this information, they were asked the following questions:

1. Is the experiment valid?
   *Options:* Yes / No / Unsure

2. Was the experiment faithfully implemented?
   *Options:* Yes / No / Unsure

3. Are the following hypotheses duplicates of the current hypothesis (see above)?
   Compare the listed hypotheses to the target hypothesis and decide whether they are exact or near-duplicates in meaning.
   *Options for each listed hypothesis:* Duplicate / Not a duplicate / Unsure

## 12  Sampling Unique Hypotheses across Methods

In Fig. 7, we aim to disentangle the efficiency of generating hypotheses that are unique (i.e., evaluating diversity) vs. our joint objective of unique surprisals (as described by our main experiments). Our results indicate that the diversity-only trend (Fig. 7) closely aligns with the unique surprisal trend across methods. In particular, MCTS results in very few duplicate hypotheses on average across datasets (low standard deviation). We conjecture that the ability to dynamically sample any node across the tree in each iteration, allows MCTS to leverage unique branches as context while prioritizing regions in the search space that are likely to result in high surprisal and diversity. Other tree search methods, such as greedy, follow instead a sequential root-to-leaf sampling procedure, which does not allow for dynamic sampling.

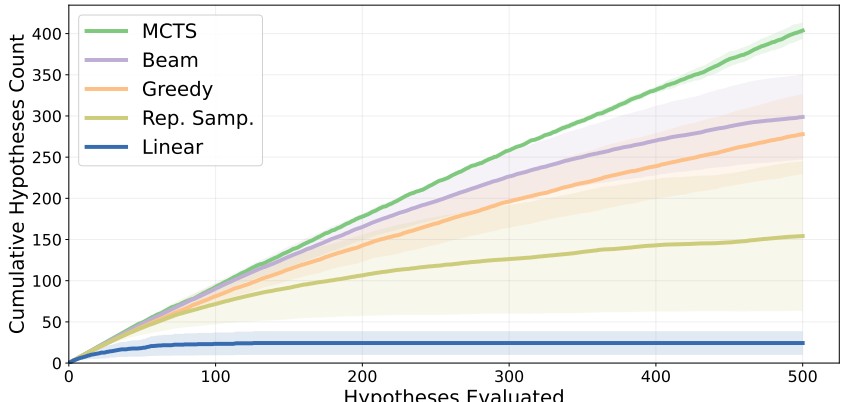

Figure 7: **Uniqueness Efficiency with GPT-4o.** Cumulative number of *unique* hypotheses discovered across timesteps within a budget of 500 evaluations, averaged over 21 datasets.

## 13  Results with o4-mini ("reasoning" models)

This section evaluates how search behavior changes when we replace GPT-4o in AUTODISCOVERY with a "reasoning" model, o4-mini, which is trained to output multiple (and longer) chains-of-thought. We repeat our main experiments with AUTODISCOVERY as well as each baseline.

**Search performance.**    Overall, we find a similar trend as in our main experiments (as shown in Fig. 8(a)) when comparing search performance across methods, with MCTS and greedy tree search outperforming other algorithms. However, we now observe no difference in performance between MCTS and greedy under our sampling budget of 500 hypotheses[7]. We conjecture that this is due to the ability of reasoning models in proposing fewer duplicate experiments. Further, Fig. 8(b) shows that, unlike with non-reasoning models, the search efficiency gradient does not measurably decline for any method but shows variable absolute values, pointing to differences in ability to find surprisals for different methods irrespective of ability to find unique hypotheses.

**o4-mini vs. GPT-4o.**    In Fig. 9(b), we plot how AUTODISCOVERY performance changes when using a reasoning model (o4-mini) versus a non-reasoning model (GPT-4o). Our results show that modest, but steady, gains can be seen in terms of cumulative surprisal counts with reasoning models. Furthermore, qualitatively, we find that the complexity of hypotheses generated by o4-mini is higher than GPT-4o. E.g., the following is a level 5 node found from the Freshwater Fish dataset (DiscoveryBench): *"Within South American freshwater-fish sub-basins, evolutionary rates (diversification and morphological evolution) exhibit significant positive spatial autocorrelation, such that geographically proximate basins have more similar rates than distant ones."*

---

[7]It is likely that we would observe a drop-off (similar to the one from the GPT-4o experiments) using greedy with a larger budget.

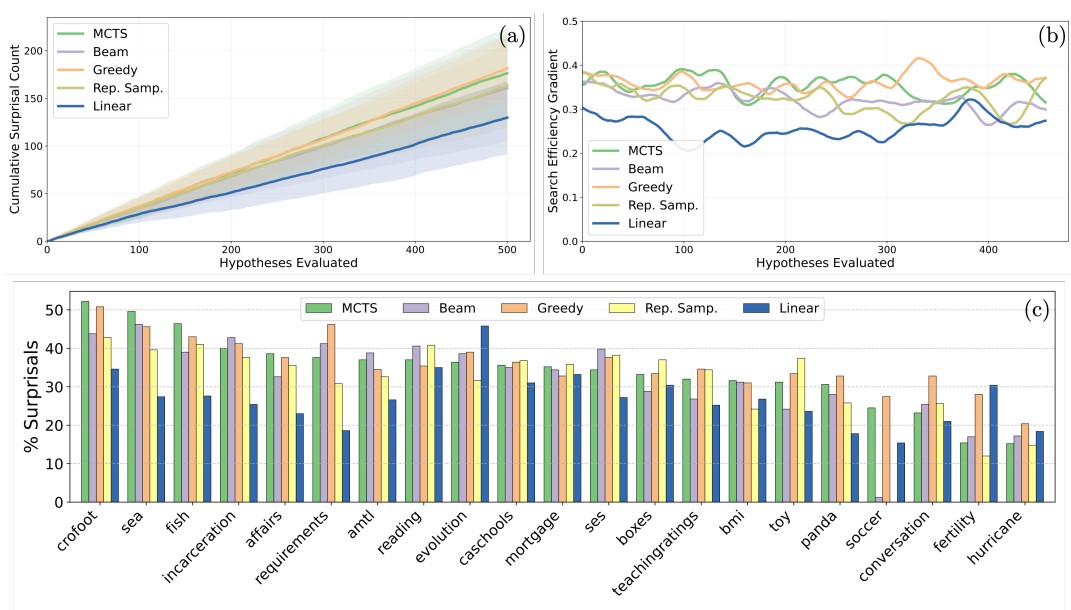

Figure 8: **Search Performance with o4-mini. (a)** Cumulative number of surprisals discovered across timesteps within a budget of 500 evaluations, averaged over 21 datasets. **(b)** Search efficiency gradient computed using a sliding window of 10 iterations. **(c)** Number of surprisals discovered per dataset.

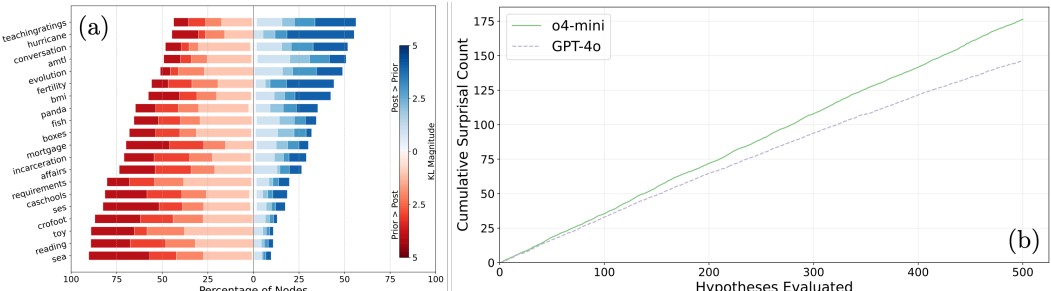

Figure 9: **(a) o4-mini: Belief shift across datasets.** Bayesian surprise under belief shift for surprisals discovered using AUTODISCOVERY, grouped by domain and direction of shift. **(b) Comparison between o4-mini and GPT-4o.** Cumulative surprisal counts using reasoning and non-reasoning models with AUTODISCOVERY (MCTS).

## 14 Example Outputs

**Code Execution:**

**Random Forest with Feature Importance**

```
exitcode: 0 (execution succeeded)
Code output: Total samples after cleaning: 84

Best parameters: {'max_depth': 5, 'min_samples_leaf': 2, 'n_estimators': 200}
AUC: mean=0.716, std=0.100
Sensitivity: mean=0.413, std=0.230
Specificity: mean=0.794, std=0.162

Feature Importances:
Age at Death         0.180995
cerad_num            0.176595
braak_num            0.148293
```

```
thal_num                0.140305
Years of education      0.135508
vascular_burden         0.126202
apoe4_count             0.051388
sex_bin                 0.040713
dtype: float64

=== Plot Analysis (1) ===
1. **Plot Type**:
   - This is a bar plot. Its purpose is to display the feature importances from a
   Random Forest model.

2. **Axes**:
   - **X-Axis**:
     - Title/Labels: Feature names such as "Age at Death," "cerad_num," etc.
     - Value Range: Categorical (not numeric).
   - **Y-Axis**:
     - Title: Importance
     - Value Range: 0.000 to 0.200

3. **Data Trends**:
   - **Bar Plot**:
     - The tallest bars indicate "Age at Death" and "cerad_num" as the most
     important features.
     - The shortest bar is "sex_bin," indicating it is the least important feature.

4. **Statistical Insights**:
   - The features have varying importance, with certain features contributing
   significantly more to the model's predictions.

5. **Annotations and Legends**:
   - Title: "Random Forest Feature Importances"
   - No additional legends or annotations are present.

6. **Overall Impression**:
   - The chart clearly indicates which features are most influential in the Random
   Forest model. This can guide further data analysis or model tuning by focusing
   on the most important features.

7. **Interpretation**:
   - "Age at Death" and "cerad_num" are critical to the model's decision-making
   process. Understanding these features' importance can provide insight into the
   modeled outcome, possibly emphasizing their relevance in predicting or
   classifying the target variable.

8. **User Objective**:
   - If the user's objective is to understand which features to prioritize in data
   collection or analysis, this plot effectively highlights the priority features
   based on their influence in the model.

9. **Limitations**:
   - Feature importances are specific to the Random Forest model and may not
   generalize across different models.
   - This plot does not provide information on feature interactions or potential
   multicollinearity.
   - The analysis doesn't take into account external factors that could influence
   the model's outcomes.
==================================================
```

## Regression

```
Code output: Interaction effects (Pathology × APOE4) on Last MMSE:
```

```
       path_var    n      coef       pval  pval_fdr  significant
0      thal_num    80 -3.691251  0.008896  0.026687         True
1   micro_gross    80  0.112921  0.768721  0.816070        False
2  micro_screen    80 -0.130590  0.816070  0.816070        False

=== Plot Analysis (1) ===
1. **Plot Type**:
   - The plot is a scatter plot with regression lines. It is used to visualize the
   relationship between two variables: thal_num and Last MMSE Score, stratified by
   APOE4 carrier status.

2. **Axes**:
   - **X-Axis**:
     - Title/Label: thal_num
     - Value Range: 0 to 5
   - **Y-Axis**:
     - Title/Label: Last MMSE Score
     - Value Range: 5 to 35

3. **Data Trends**:
   - The scatter plot shows different linear trends for APOE4 carriers (1) and
   non-carriers (0).
   - Carriers (blue) show a steeper declining trend compared to non-carriers (red).
   - There is some overlap between the data, but darker regions indicate areas with
   more points.

4. **Statistical Insights**:
   - The regression lines suggest that APOE4 carriers tend to have lower MMSE
   scores as thal_num increases, more so than non-carriers.

5. **Annotations and Legends**:
   - The legend indicates two groups: APOE4 carriers (1) and non-carriers (0),
   represented by different colors (blue for carriers, red for non-carriers).

6. **Overall Impression**:
   - The analysis indicates that as the thal_num increases, the MMSE score tends to
   decrease, especially for APOE4 carriers. This suggests a potential impact of
   thal_num on cognitive performance, moderated by APOE4 carrier status.

7. **Interpretation**:
   - The data implies that individuals carrying the APOE4 gene may experience a
   more pronounced decline in cognitive performance (as measured by the MMSE score)
   associated with increasing thal_num.

8. **User Objective**:
   - If the user's objective is to assess cognitive decline in relation to thal_num
   and genetic factors, the plot provides a clear visual representation of these
   relationships.

9. **Limitations**:
   - The data does not indicate causality.
   - There might be confounding factors not accounted for in the analysis.
   - The sample size at each thal_num level and the variability are not specified,
   which may affect the reliability of the trend lines.
==================================================
```

## Ordinary Least Squares Regression

```
Regression Summary:
                        OLS Regression Results
==============================================================================
Dep. Variable:              last_mmse   R-squared:                       0.241
```

```
Model:                              OLS   Adj. R-squared:                    0.189
Method:                   Least Squares   F-statistic:                       4.690
Date:                 Tue, 20 May 2025   Prob (F-statistic):             0.000895
Time:                         12:10:18   Log-Likelihood:                   -221.68
No. Observations:                   80   AIC:                                455.4
Df Residuals:                       74   BIC:                                469.7
Df Model:                            5
Covariance Type:             nonrobust
===============================================================================
=====
                        coef    std err          t      P>t      [0.025
                     0.975]
-------------------------------------------------------------------------------
-----
Intercept            20.4041     14.935      1.366      0.176      -9.354
50.162
C(sex)[T.Male]        0.1244      0.961      0.129      0.897      -1.791
2.039
years_edu            -0.0143      0.797     -0.018      0.986      -1.602
1.573
braak_num            -1.5258      2.725     -0.560      0.577      -6.956
3.905
years_edu:braak_num  -0.0050      0.166     -0.030      0.976      -0.336
0.326
age_at_death          0.1295      0.063      2.043      0.045       0.003
0.256
===============================================================================
Omnibus:                         8.277   Durbin-Watson:                     2.013
Prob(Omnibus):                   0.016   Jarque-Bera (JB):                  8.367
Skew:                           -0.590   Prob(JB):                         0.0152
Kurtosis:                        4.058   Cond. No.                       3.95e+03
===============================================================================

Notes:
[1] Standard Errors assume that the covariance matrix of the errors is correctly
specified.
[2] The condition number is large, 3.95e+03. This might indicate that there are
strong multicollinearity or other numerical problems.

Interaction term 'years_edu:braak_num': coefficient = -0.0050, p-value = 0.9759
```

**Clustering**

```
    Code output:
=== Plot Analysis (1) ===
1. **Plot Type**:
   - This is a dendrogram, used in hierarchical clustering to show the arrangement
   of clusters formed by the algorithm.

2. **Axes**:
   - **X-axis**: Labeled as "Donors" with no specific units indicated. It
   represents the data points (or samples) being clustered.
   - **Y-axis**: Labeled as "Distance." This axis shows the distance or
   dissimilarity between clusters. The value range is from 0.0 to 20.0.

3. **Data Trends**:
   - The dendrogram shows the hierarchical relationship of the clusters.
   - Longer vertical lines at the top indicate greater dissimilarity between
   clusters.
   - Shorter lines at the bottom represent more closely related clusters.

4. **Statistical Insights**:
```

- The height at which two clusters join indicates the dissimilarity. Higher
      joins mean more dissimilar clusters.

5. **Annotations and Legends**:
   - No specific annotations or legends are present. The branching structure itself
   provides the clustering information.

6. **Overall Impression**:
   - The data points are grouped hierarchically, revealing patterns of similarity.
   The clusters closer to the bottom are more similar, whereas those joining higher
   have more dissimilarity.

7. **Interpretation**:
   - The dendrogram allows visualization of the data's natural groupings, which can
   be useful for determining the optimal number of clusters by cutting the
   dendrogram at different heights.

8. **User Objective**:
   - Likely to identify the grouping structure of the donors based on certain
   attributes or metrics of similarity/distance.

9. **Limitations**:
   - Dendrograms can become cluttered with large datasets, making them harder to
   interpret.
   - The method's sensitivity to distance metrics and linkage methods can affect
   conclusions significantly.
==================================================

=== Plot Analysis (1) ===
1. **Plot Type:**
   - **Type:** Heatmap
   - **Purpose:** To visually represent the normalized values of different
   neuropathological features, allowing for easy identification of patterns,
   variations, and extreme values across various categories.

2. **Axes:**
   - **Titles and Labels:**
     - **X-Axis:** Neuropathological features such as
     Overall_AD_neuropathological_Change, Thal, Braak, etc.
     - **Y-Axis:** Sample or observation index (not labeled explicitly, appears
     categorical).
   - **Value Ranges:**
     - **X-Axis:** Categorical features.
     - **Y-Axis:** Categorical index.

3. **Data Trends:**
   - **High Values (Red):** Notable in columns such as "LATE" and
   "Arteriosclerosis."
   - **Low Values (Blue):** Prominent in "Overall_AD_neuropathological_Change" and
   "Total_Microinfarcts."
   - **Patterns:** Stripes of consistent color indicate similarities in features
   across samples.

4. **Statistical Insights:**
   - **Normalized Values:** Range from -2.0 (low) to 2.0 (high), enabling
   comparison across features.
   - **Distribution:** Variation suggests differences in presence and severity of
   the conditions measured.

5. **Annotations and Legends:**
   - **Legend:** Color bar on the right labeling the normalized values from -2.0
   (blue) to 2.0 (red).

6. **Overall Impression:**
   - The heatmap effectively shows diverse variability in neuropathological measures. Features such as "Arteriosclerosis" and "LATE" display higher normalized values, suggesting greater severity in those areas for certain individual observations.

7. **Interpretation:**
   - **Conclusions:** Some features, like "Overall_AD_neuropathological_Change," often exhibit low values across samples, indicating a potential pattern of less severity or frequency in this dataset compared to features like "Arteriosclerosis."

8. **User Objective:**
   - **Objective:** Identify patterns in neuropathological data that may correspond to different pathological states or severities across samples. Recognize high and low prevalence of pathological features.

9. **Limitations:**
   - **Data Bias:** The heatmap portrays normalized data; without raw data, assessing actual severity is difficult.
   - **Interpretation Bias:** Over-reliance on color may overlook nuanced details.
   - **Sample Size:** Not visible; the sample size could affect the robustness of observed patterns.

This heatmap provides a comprehensive visualization of various neuropathological features, offering insights into potential underlying patterns across samples. For further analysis, exploring how these features correlate with clinical outcomes or demographic data could be valuable.
==================================================
Cluster 1 (n=14):
  Mean Age at Death: 88.21
  Mean RIN: 8.58
  Sex distribution: {'Female': 0.5714285714285714, 'Male': 0.42857142857142855}
  Cognitive Status distribution: {'No dementia': 0.8571428571428571, 'Dementia': 0.14285714285714285}

Cluster 2 (n=17):
  Mean Age at Death: 90.00
  Mean RIN: 8.40
  Sex distribution: {'Female': 0.5294117647058824, 'Male': 0.47058823529411764}
  Cognitive Status distribution: {'No dementia': 0.6470588235294118, 'Dementia': 0.35294117647058826}

Cluster 3 (n=5):
  Mean Age at Death: 90.40
  Mean RIN: 8.47
  Sex distribution: {'Female': 0.6, 'Male': 0.4}
  Cognitive Status distribution: {'Dementia': 0.6, 'No dementia': 0.4}

Cluster 4 (n=19):
  Mean Age at Death: 89.47
  Mean RIN: 8.65
  Sex distribution: {'Female': 0.631578947368421, 'Male': 0.3684210526315789}
  Cognitive Status distribution: {'Dementia': 0.5263157894736842, 'No dementia': 0.47368421052631576}

Cluster 5 (n=29):
  Mean Age at Death: 87.45
  Mean RIN: 7.51
  Sex distribution: {'Female': 0.6551724137931034, 'Male': 0.3448275862068966}
  Cognitive Status distribution: {'Dementia': 0.7241379310344828, 'No dementia': 0.27586206896551724}

## 15 Example Errors

**Excessive Code Execution Output.** Automatic code generation is susceptible to unexpected errors, e.g., due to malformed arguments. Such instances, therefore, may result in uncaught exceptions and repetitions. For example, the following example shows the same log message being generated >66,000 times for a single hypothesis node.

```
[LightGBM] [Warning] No further splits with positive gain, best gain: -inf
[LightGBM] [Warning] No further splits with positive gain, best gain: -inf
...
```

**Network Error.** Our current experiments use OpenAI API calls and are, thus, reliant on the stability of their hosted service. Future work may look into using local models.

```
Traceback (most recent call last):
...
  File "/lib/python3.11/site-packages/autogen/oai/client.py", line 466, in
  _create_or_parse
    return self._oai_client.chat.completions.create(*args, **kwargs)
           ~~~~~~~~~~~~~~~~~~~~~~~~~~~~~~~~~~~~~~~~~~~~~~~~~~~~~~~~~~~
  File "/lib/python3.11/site-packages/openai/_utils/_utils.py", line 287, in wrapper
    return func(*args, **kwargs)
           ~~~~~~~~~~~~~~~~~~~~~~
  File
  "/lib/python3.11/site-packages/openai/resources/chat/completions/completions.py",
  line 925, in create
    return self._post(
           ~~~~~~~~~~~
  File "/lib/python3.11/site-packages/openai/_base_client.py", line 1239, in post
    return cast(ResponseT, self.request(cast_to, opts, stream=stream,
    stream_cls=stream_cls))
                           ~~~~~~~~~~~~~~~~~~~~~~~~~~~~~~~~~~~~~~~~~~~~~~~~~~~~~~~~~~~~ ⌋
                           ~~~~~~~~~~
  File "/lib/python3.11/site-packages/openai/_base_client.py", line 1001, in request
    raise APIConnectionError(request=request) from err
```

**Flagged Prompts.** In a few instances, we observe that API calls are rejected as violating usage policy when we're processing datasets that involve race or gender, likely the effect of aggressive safety tuning.

```
...
=== Plot Analysis (1) ===
1. **Plot Type**:
   - This is a heatmap.
   - Purpose: To visualize the relationship between age, culture, and conformity to
   majority proportions.

2. **Axes**:
   - **X-axis**: Labeled "Culture" with values 1 to 8.
   - **Y-axis**: Labeled "Age" with values 4 to 14.
   - No specific units are provided for these axes.

3. **Data Trends**:
   - Areas of high values (yellow) indicate high proportions of
   majority-conformity, such as age 12 and culture 1, and ages 13-14 with cultures 7
   and 8.
   - Areas of low values (dark blue) indicate low conformity, noticeable at certain
   combinations like ages 9 and 11 with culture 8.
```

4. **Statistical Insights**:
   - Values range from 0.0 to 1.0, representing proportions.
   - High conformity (values of 1.0) is clearly marked, suggesting strong
   tendencies toward majority-conformity in specific groups.

5. **Annotations and Legends**:
   - The heatmap includes a color bar on the right side to indicate the proportion
   range (0.0 to 1.0).
   - Numerical values within the heatmap provide specific data points for
   conformity proportions.

6. **Overall Impression**:
   - There are distinct clusters of high conformity at specific age and culture
   intersections.
   - There is variability across both axes.

7. **Interpretation**:
   - Certain age groups show strong conformity in specific cultures, which could
   indicate social or cultural influences.
   - The patterns suggest that age and cultural background significantly impact
   conformity behaviors.

8. **User Objective**:
   - If the objective is to understand how culture and age influence conformity,
   this heatmap effectively highlights potential areas for further sociological or
   psychological analysis.

9. **Limitations**:
   - The heatmap does not indicate causality.
   - Missing data points could lead to incomplete conclusions.
   - Cultural categories and age groupings might be subjective or oversimplified,
   impacting data interpretation.

=================================================
...
openai.BadRequestError: Error code: 400 - {'error': {'message': 'Invalid prompt:
your prompt was flagged as potentially violating our usage policy. Please try again
with a different prompt:
https://platform.openai.com/docs/guides/reasoning#advice-on-prompting', 'type':
'invalid_request_error', 'param': No
ne, 'code': 'invalid_prompt'}}
```

**Timeout.** The discovery agent, in some cases, may result in timeout errors. This is particularly seen when the generated experiment program requires a long-running operation or the amount of data being processed is large.

```
{"code":"import subprocess, sys\n# Install SALib for sensitivity analysis\ndef
install(pkg): subprocess.check_call([sys.executable, '-m', 'pip', 'install',
'--quiet', pkg])\ninstall('SALib')\n\nimport pandas as pd\nimport numpy as
np\nimport matplotlib.pyplot as plt\nfrom sklearn.preprocessing import Stan
dardScaler\nfrom sklearn.gaussian_process import GaussianProcessRegressor\nfrom
sklearn.gaussian_process.kernels import RBF, WhiteKernel\nfrom SALib.sample import
saltelli\nfrom SALib.analyze import sobol\nimport
warnings\nwarnings.filterwarnings('ignore')\n\n# 1. Data Preparation\ndf =
pd.read_csv('nls_b
mi_processed.csv')\n# Compute raw saving direction and age
squared\ndf['SAVING_DIR_raw'] = np.where(df['DISSAVED']==1, -1,\n
np.where(df['SAMESAVE']==1, 0, 1))\ndf['AGE2'] = df['AGE']**2\n# Standardize
continuous variables\ncont = ['AGE','INCOME','BMI','SAVING_DIR_raw']\nsc
```

```
aler = StandardScaler()\ndf_z = scaler.fit_transform(df[cont])\nfor i, col in
enumerate(cont): df[f'{col}_z'] = df_z[:,i]\n# Ensure binary
covariates\ndf['GENDER_MALE'] = (df['GENDER']=='MALE').astype(int)\ndf['BLACK'] =
df['BLACK'].astype(int)\ndf['HISPANIC'] = df['HISPANIC'].astype(int)\n# Predictor
nam
es\nelements =
['AGE_z','INCOME_z','BMI_z','SAVING_DIR_raw_z','GENDER_MALE','BLACK','HISPANIC']\nX
= df[elements].values\nY = df['BMI'].values\nn, p = X.shape\n\n# 2. Fit Gaussian
Process surrogate on full data\ngp = GaussianProcessRegressor(kernel=RBF(length_sc
ale=np.ones(p))+WhiteKernel(noise_level=1.0)
,\n                            normalize_y=True, random_state=0).fit(X, Y)\n\n#
3. Sobol sampling problem definition\ndescriptor = {\n    'num_vars': p,\n
'names': elements,\n    'bounds': [[df[col+'_z'].min(), df[col+'_z'].max()] for col
in cont]\n}\n# Use smaller sample for speed\nN = 200\nparam_va
lues = saltelli.sample(descriptor, N, calc_second_order=False)\n\n# 4. Evaluate
surrogate and compute Sobol indices\ny_sobol = gp.predict(param_values)\nsi =
sobol.analyze(descriptor, y_sobol, calc_second_order=False,
print_to_console=False)\nS1 = si['S1']; ST = si['ST']\n\n# 5. Bootstrap uncertainty
esti
mation (B=100)\nB = 100\nboot_S1 = np.zeros((B, p))\nboot_ST = np.zeros((B,
p))\nfor b in range(B):\n    idx = np.random.choice(n, n, replace=True)\n    Xb, Yb
= X[idx], Y[idx]\n    gp_b = GaussianProcessRegressor(kernel=RBF(length_scale=np.o
nes(p))+WhiteKernel(noise_level=1.0),\n
        normalize_y=True, random_state=0).fit(Xb, Yb)\n    ys =
        gp_b.predict(param_values)\n    sib = sobol.analyze(descriptor, ys,
        calc_second_order=False, print_to_console=False)\n    boot_S1[b,:] =
        sib['S1']\n    boot_ST[b,:] = sib['ST']\nci_S1 = np.percentile(boot_S1,
        [2.5,97.5], axis=0)\nci_ST = np
.percentile(boot_ST, [2.5,97.5], axis=0)\n\n# 6. Visualization of Sobol
indices\nplt.figure(figsize=(8,4))\nindices = np.arange(p)\nwidth =
0.4\nplt.bar(indices - width/2, S1, width, yerr=[S1-ci_S1[0], ci_S1[1]-S1],
capsize=5, label='First-order')\nplt.bar(indices + width/2, ST, width,
yerr=[ST-ci_ST[0],
ci_ST[1]-ST], capsize=5, label='Total-order')\nplt.xticks(indices, elements,
rotation=45, ha='right')\nplt.ylabel('Sobol index')\nplt.title('Sobol Sensitivity
Indices for BMI GP Surrogate')\nplt.legend()\nplt.tight_layout()\nplt.show()\n\n#
7. Reporting results\nprint(\"Sobol Sensitivity Indices with 95%
CIs:\")\nfor i, var in enumerate(elements):\n    print(f\"{var}: S1 = {S1[i]:.3f}
[{ci_S1[0,i]:.3f}, {ci_S1[1,i]:.3f}], \",\n          f\"ST = {ST[i]:.3f}
[{ci_ST[0,i]:.3f}, {ci_ST[1,i]:.3f}]\")"}
```

--------------------------------------------------------------------------------

Next speaker: code_executor

>>>>>>>> EXECUTING CODE BLOCK (inferred language is python)...
code_executor (to chat_manager):

exitcode: 124 (execution failed)
Code output:
Timeout

--------------------------------------------------------------------------------

Next speaker: experiment_analyst

{"error":true,"analysis":"The Sobol sensitivity analysis timed out during bootstrap
computation (B=100). The high computational cost of refitting the Gaussian-process
surrogate and recalculating Sobol indices for each replicate exceeded the time
limit. Consider reducing the sample size for Sobol sampling
(N) and the number of bootstrap replicates (B), or implementing parallelization to
complete the analysis."}

