# OpenReview forum: "AutoDiscovery: Open-ended Scientific Discovery via Bayesian Surprise"
_NeurIPS.cc/2025/Conference — NeurIPS 2025 poster_

### Official Review · Reviewer_TsGq · 2025-06-20

**Clarity:** 3
**Significance:** 3
**Originality:** 3
**Rating:** 5
**Confidence:** 4

**Summary:**

The authors observed that much of the prior work on Autonomous Scientific Discovery (ASD) has been goal-driven. In contrast, they aimed to propose a hypothesis generation method for a more open-ended setting, where the ASD system explores more broadly by generating hypotheses based on its own measures of research promise, executing the aforementioned steps, and then using the results to propose new hypotheses in a never-ending process.

To achieve this, the authors proposed a method that uses Monte Carlo Tree Search (MCTS) to explore hypotheses that yield high Bayesian surprise. Specifically, they conceptualized a tree where each node represents a hypothesis. For each node, they defined a hypothesis as surprising if it significantly changes the beliefs of a large language model (LLM) before and after observing the experimental result. Based on this formulation, they used MCTS to search for such hypotheses.

They evaluated their method on several datasets and demonstrated that it could discover more surprising hypotheses than baseline methods. Moreover, the surprises identified by their system were shown to correlate with the surprises perceived by human researchers.

**Questions:**

- Let me confirm the definition of a hypothesis. How is ‘hypothesis’ defined here? Are you referring to experimental ideas as hypotheses?
- Is there any qualitative evaluation or explanation of why humans made such judgments? If so, I would like to refer to it.

**Ethical Concerns:**

["NO or VERY MINOR ethics concerns only"]

**Final Justification:**

In my initial evaluation, I was concerned that although this study conducts a certain level of sound validation, the comparison with existing hypothesis generation and exploration methods is limited. However, after receiving the authors’ response, I came to interpret that the main contributions—the proposed exploration reward and framework—are supported by experiments. Additionally, the authors conducted an analysis showing that the exploration framework remains effective to some extent even when the internal hypothesis generation method is varied. This helped me judge that the experiments appropriately demonstrate the effectiveness of the exploration method.

**Limitations:**

Since the study aims for a more autonomous framework, I believe the practical quality of the generated output has declined, which I consider to be a limitation.

Also, the assumption of the existence of a dataset can be seen as a limitation in light of the ultimate goal this research is aiming to achieve.

**Quality:**

3

**Strengths And Weaknesses:**

Improving autonomy and open-endedness in ASD (Autonomous Scientific Discovery) is a central research challenge. The authors’ work is notable in this context, as they explore methodologies for autonomously evaluating hypotheses and proposing subsequent ones.

While the idea of using Bayesian beliefs for hypothesis evaluation has been long-standing, the authors provide concrete computational formulations. Moreover, their method is carefully designed—not only to maximize Bayesian surprise, but also to ensure that such surprise leads to meaningful shifts, demonstrating thoughtful engineering.

The experiments are well-validated using multiple datasets and include human evaluations, adding robustness to the claims. Additionally, the authors conduct a variety of analytical experiments to examine their method from different perspectives, which further broadens the implications of this research. The paper documents the experimental details—including prompts, annotation processes, and algorithmic explanations—in both the main text and appendices, ensuring transparency in the verification process.

The paper also appropriately situates itself within the context of prior work. Not only does it reference background research relevant to this study, but it also draws upon a wide range of earlier findings in developing its methodology, making it a fruitful reference for future researchers developing novel approaches in this field.

That said, there are several concerns that warrant discussion.

First is the practical significance of improving open-endedness, which this work aims to address. As I understand it, the authors recognize that many previous efforts in ASD have assumed the availability of data and goals (as well as researcher questions), and seek to move beyond goal-driven constraints by allowing the system to autonomously determine its own goals—thus achieving a more open-ended hypothesis generation.

I agree that enabling AI to autonomously define its own goals is essential for autonomous scientific discovery. Of course, no goal can be defined from a blank slate—some prior assumption is always necessary. The authors assume that only data (and metadata) are given, and investigate how a system can autonomously discover goals from that. However, the importance of this “data-only” assumption compared to other initializations remains debatable. For example, instead of specifying a concrete research goal (which may be overly constraining), one could provide only an abstract goal (e.g., “develop safe AI”) and allow the system to autonomously derive more concrete research problems. This may still be more open-ended than traditional goal-driven systems.

In this alternative framework, the choice of dataset is not predetermined; in fact, the AI could autonomously select or even create its own datasets from the web. This arguably introduces greater open-endedness than assuming a fixed dataset. For instance, systems such as MLR-Copilot (Li et al., 2024) and The AI Scientist-v2 (Yamada et al., 2025), while not fully automating dataset acquisition, allow autonomous selection among diverse datasets by leveraging collections like Huggingface—similar to Prompt2Model (Viswanathan et al., 2023).

Moreover, as the authors themselves mention, one could envision a framework where research problems are selected based on a literature survey, even in the absence of explicit goals. While the authors argue that their approach allows exploration of a broader space compared to literature-based methods, it’s worth noting that assuming only a literature survey grants flexibility in dataset selection—something the present approach lacks.

Considering these perspectives, it is not immediately obvious which framework—assuming only an abstract goal, a literature survey, or a dataset—leads to greater open-endedness in terms of goal selection. In real-world research scenarios, starting from a high-level objective and designing a study to achieve it may have more practical impact than starting from data alone. I acknowledge that this may reflect my own bias, and I would be very interested in hearing the authors’ reasoning on why their “data-only” approach is particularly important for advancing open-endedness.

Second, regarding related work: one closely relevant study is data-to-paper by Ifargan et al. (2024), which demonstrates fully autonomous LLM-driven research from raw data to publishable papers. While this study differs significantly in its methodology and does not diminish the novelty of the present work, it should still be acknowledged and discussed as a related approach. Also, this research seems to be related to the automation of data analysis. Since there are many previous studies on this topic as well, it would be good to refer to them appropriately.

Next, regarding the experiments: Table 1 shows that the system can indeed generate hypotheses that are surprising to humans. This seems sufficient for validating the main claim. However, given the existence of many previous works on hypothesis generation, it would strengthen the case if the proposed method were compared—even partially—to some of these prior approaches. Although perfectly fair comparisons may be difficult due to differing assumptions, such a comparison would help substantiate the practical value of this method. Looking at the generated samples, it appears that the code produced involves relatively simple data analysis.

Finally, let me discuss the output from the system. At first glance, it seems challenging to derive hypotheses with academic value from such output. I was curious about the qualitative evaluation—specifically, in what sense the generated hypotheses were perceived as “surprising.” While these hypotheses might indeed be more surprising compared to a baseline, I also wondered how they qualitatively compare to hypotheses generated by other hypothesis-generation methods. For instance, if we naively sampled hypotheses at random from both groups and asked an LLM-as-Judge to evaluate which hypothesis seemed better, I would be interested in knowing to what extent one group would be judged statistically superior.

There are some concerns, but overall, I believe the strengths of this work outweigh the weaknesses. I recommend acceptance of this paper.

---

> ### Author Rebuttal · Authors · 2025-07-31
>
> Thank you for taking the time to review our work! We address your concerns and questions below.
>
> ---
>
> > `W1: …it is not immediately obvious which framework—assuming only an abstract goal, a literature survey, or a dataset—leads to greater open-endedness…I would be very interested in hearing the authors’ reasoning on why their “data-only” approach is particularly important for advancing open-endedness.`
>
> We emphasize that our main contribution is in presenting an efficient computational framework for repeatedly sampling hypotheses that are likely to lead to scientific discoveries (AutoDS). Our specific choice of using data-driven discovery as a test-bed was motivated by the feasibility of data-driven hypotheses to be evaluated via statistical tests using python programs. In particular, this paradigm does not need to access external data or perform physical experiments to verify hypotheses. Therefore, indeed, the search framework of AutoDS can be utilized with just an abstract goal or with literature. However, the mechanism for hypothesis verification would need to be specified.
>
> ---
>
> > `W2: regarding related work: one closely relevant study is data-to-paper by Ifargan et al. (2024), which demonstrates fully autonomous LLM-driven research from raw data to publishable papers. While this study differs significantly…it should still be acknowledged and discussed as a related approach…`
>
> Thank you for the reference! We will surely add it to our related works discussion in the final paper.
>
> ---
>
> > `W3: …Table 1 shows that the system can indeed generate hypotheses that are surprising to humans…it would strengthen the case if the proposed method were compared—even partially—to some of these prior approaches…Looking at the generated samples, it appears that the code produced involves relatively simple data analysis.`
>
> > `W4: …I also wondered how they qualitatively compare to hypotheses generated by other hypothesis-generation methods…`
>
> As mentioned above, our main contribution is in presenting a search framework. The inner-loop of hypothesis generation used in AutoDS can easily be swapped out for a different method. For the instantiation in our experiments, we use DataVoyager (Majumder et al., 2024) for the inner-loop, but could have instead used a literature-based method like Scideator (Radensky et al., 2024) as well. Furthermore, as described in our supplementary material L177-180, the statistical analyses deeper in the tree, especially with o4-mini, show increased complexity. E.g., *“Within South American freshwater-fish sub-basins, evolutionary rates (diversification and morphological evolution) exhibit significant positive spatial autocorrelation, such that geographically proximate basins have more similar rates than distant ones.”*
>
> **References:**
> *Data-driven Discovery with Large Generative Models. Majumder et al., 2024.
> Scideator: Human-LLM Scientific Idea Generation Grounded in Research-Paper Facet Recombination. Radensky et al., 2024.*
>
> ---
>
> > `Q1: Let me confirm the definition of a hypothesis. How is ‘hypothesis’ defined here? Are you referring to experimental ideas as hypotheses?`
>
> Please see our formal definition in Section 2 Preliminaries L85-89, which follows from Majumder et al., 2025. We clarify that we use an “experiment” as a verification procedure to determine whether the hypothesis is supported or not given a dataset (in the data-driven discovery paradigm).
>
> **References:**
> *DiscoveryBench: Towards Data-Driven Discovery with Large Language Models. Majumder et al., 2025.*
>
> ---
>
> > `Q2: Is there any qualitative evaluation or explanation of why humans made such judgments? If so, I would like to refer to it.`
>
> We do not collect reasoning from annotators about their beliefs and, in fact, note that it may be hard to verbalize the reason when prompted to provide a prior belief given a hypothesis (Rozenblit & Keil, 2002). We do note, however, that our annotations are averaged over 3 annotator responses, and, therefore, aim to capture variability in responses.
>
> **References:**
> *The misunderstood limits of folk science: an illusion of explanatory depth. Rozenblit & Keil, 2002.*
>
> ---
>
> > `Limitations: Since the study aims for a more autonomous framework, I believe the practical quality of the generated output has declined, which I consider to be a limitation. Also, the assumption of the existence of a dataset can be seen as a limitation in light of the ultimate goal this research is aiming to achieve.`
>
> We re-emphasize that the main contribution of our work is in providing a computational search framework for scientific discovery. The quality of the inner-loop, therefore, is a related but distinct question. In order to improve the robustness of our results, one may, for example, use a mechanism such as POPPER to conduct several rounds of falsification experiments before concluding in order to build confidence in the verification of a proposed hypothesis. Similarly, one may choose to augment an LLM with relevant literature or to use models fine-tuned on domain-specific knowledge to inform both hypothesis generation and belief elicitation. AutoDS can seamlessly integrate each of these modifications.

---

> > ### Comment · Reviewer_TsGq · 2025-08-05
> > **Response to authors**
> >
> > Thank you for your response. Here is my comment:
> >
> > W1
> > There was a slight misunderstanding on my part regarding what the authors refer to as “open-ended.” Thank you for the clarification.
> >
> > W2
> > Thank you. I now understand that Table 1 compares different automatic reward signals rather than evaluating the entire hypothesis generation approach. In the Introduction, you mention that prior studies have used rewards such as “diversity and LLM-as-judge proxies for human interestingness, novelty, or utility [Zhang et al., 2023a; Lu et al., 2024].” Have you included any comparisons with novelty or diversity?
> >
> > Q1
> > I had overlooked that—thank you for the clarification.
> >
> > Q2
> > I understand that collecting annotators’ reasoning was deemed difficult to verbalize and therefore not conducted. Thank you for the explanation.
> >
> > Limitation
> > I now understand that the primary focus is on the search component and that it is considered separate from the inner-loop. However, since the inner-loop and the search are ultimately executed together, I believe the performance of one can inevitably affect the other. At the very least, when evaluating the overall system, it is not necessarily obvious that the influence of the two components can be completely disentangled. For example, hypothetically, if a hypothesis generation method consistently produces similar hypotheses in terms of Bayesian surprise due to a certain bias, the search process may not function efficiently. Therefore, if the claim is that the proposed search framework is effective independently of the inner-loop, I believe it is necessary to show that it works effectively across multiple, diverse inner-loops as a controlled condition. This is especially true when the evaluation focuses on the generated hypotheses themselves rather than just the efficiency of the search process.
> > If you have already conducted any robustness checks using different hypothesis generation methods (i.e., different inner-loops) and I happened to miss them, I would appreciate it if you could kindly point them out.

---

> > > ### Author Response · Authors · 2025-08-05
> > >
> > > We're glad that our rebuttal has addressed some of your comments and questions!
> > >
> > > ---
> > >
> > > > `...I now understand that the primary focus is on the search component and that it is considered separate from the inner-loop...it is not necessarily obvious that the influence of the two components can be completely disentangled...necessary to show that it works effectively across multiple, diverse inner-loops as a controlled condition...If you have already conducted any robustness checks using different hypothesis generation methods (i.e., different inner-loops) and I happened to miss them, I would appreciate it if you could kindly point them out.`
> > >
> > > We agree that the inner-loop of hypothesis generation/verification is inextricably linked to the outer-loop of search. On your suggestion, below we show results from a new set of AutoDS runs with 3 variants of our inner-loop – "original", "research query"-driven, and "literature"-driven – which differ in the conditioning that is provided for hypothesis generation in the inner-loop. We report results on the Seattle Alzheimer's (SEA-AD) dataset for a curtailed run of 75 experiments.
> > >
> > > **Quantitative results:**
> > > | Inner-Loop         | Surprisal Proportion |
> > > |----------------|----------------------|
> > > | Original       | 0.62                |
> > > | Research query | 0.44                |
> > > | Literature          | 0.64                |
> > >
> > > **Qualitative examples:**
> > > - Original: *Local tissue cellularity (Hematoxylin number per area) across laminar bins (Layers 1–4 vs Layers 5–6) correlates positively with layer-specific astrocyte activation (GFAP percent area in the same bins).*
> > > - Research query: *In donors with ‘High’ Overall AD neuropathological change, the average size of amyloid-β plaques (average 6E10 object area) increases progressively from superficial layers (1–4) to deep layers (5–6).*
> > > - Paper: *Laminar tau gradient does not significantly improve prediction of clinical Alzheimer’s diagnosis beyond Braak stage in the SEA-AD cohort.*
> > >
> > > Another way to alter the inner-loop may be to change the model used. In our supplementary material, we show this comparison between GPT-4o and o4-mini (“Results with o4-mini (“reasoning” models)” and Figures 5 and 6). We found that the same trend across search methods holds for both models.
> > >
> > > Given the above examples and analyses, we, thus, believe that AutoDS does demonstrate the ability to operate with several choices of the inner-loop. We, nonetheless, agree that several methodological advances may be made in the AutoDS inner-loop related to how hypotheses are proposed, refined, critiqued, and verified, as is already being done in the community.
> > >
> > > We hope this addresses your concern and we will surely add this new analysis to the final version of the paper.

---

> > > > ### Comment · Reviewer_TsGq · 2025-08-06
> > > > **Response to authors**
> > > >
> > > > Thank you for your response. I see that the method is expected to work reasonably well even with different inner loops. I appreciate your sincere and thoughtful engagement with many of my concerns. As many of concerns have been addressed to a satisfactory extent, I would like to raise my score. Thank you.

---

> > > > > ### Author Response · Authors · 2025-08-06
> > > > >
> > > > > We truly appreciate it! Thank you again for engaging in a constructive dialogue and for the helpful suggestion!

---

### Official Review · Reviewer_7cwK · 2025-06-27

**Clarity:** 4
**Significance:** 2
**Originality:** 2
**Rating:** 4
**Confidence:** 4

**Summary:**

This paper is about LLM-based autonomous scientific discovery. It focuses on open-ended hypotheses selection and proposes to use bayesian surprise over diversity and human interestingness as an objective to be optimized. Bayesian surprise is calculated by assuming that a hypothesis is either true of false (no graded correctness) with respect to data and a verifier, based on a given data, can be synthesized and executed. To optimize this quantity, the paper uses MCTS to search for hypotheses with high surprisal (which is Bayesian Surprise under belief shift). Experiments are done on a subset of DiscoveryBench, BLADE, and SEA-AD, totaling 21 datasets. Results show that MCTS outperforms other methods in optimizing this bayesian surprisal objective and demonstrate on 4 datasets that humans (3 MS/PhDs) hypotheses chosen by the proposed method more surprising than MCTS optimizing other objectives.

**Questions:**

- How would the proposed method work with a hypothesis like "the coin is unbiased" and observe coin flips?
- Displaying more qualitative results on how the proposed method is doing compared to other baselines / ablated methods would be good for the paper.
- Human evaluation is done on the hypotheses generated on only 4 out of 21 datasets. What are these 4 datasets? What do they look like?
- What do the error bars look like for table 1, given that there are only 3 human evaluators? How was inter-annotator agreement calculated? Are the human evaluators the authors themselves? Did the evaluators know which method was used to generate each hypothesis? I think the paper can benefit greatly from a full, proper human study

**Ethical Concerns:**

["NO or VERY MINOR ethics concerns only"]

**Final Justification:**

All of my concerns are addressed.

Other things that prevent me from giving even higher scores include the complexity of the domains used (also related to w1a) (they are not simple, and they are a good start, but they are not particularly impressive).

**Limitations:**

yes

**Quality:**

2

**Strengths And Weaknesses:**

Strength:
- The proposed method is well motivated (surprise, progressive widening, deduplication)
- The paper is well written, easy to read and follow
- The paper shows that humans do find hypotheses generated by the proposed method surprising, and I like how the paper completes this part of the story by showing that interesting ness and utility lack clear semantics, as judged by humans.

Weakness:
- I find the results to be very weak in supporting the main message of the paper:
The paper is proposing a new objective to optimize ASD, and I do not see how section 5.1 (whose main message is MCTS outperforms other search strategies) contribute to the main story: given an objective, it's not surprising or novel that MCTS performs better than beam search, greedy search etc. In section 5.2, only 3 human participants participate in evaluation. No error bars are provided. And given how LLM-heavy the method is, I expect evaluations on more LLMs, not just a single one (gpt-4o).
- The assumptions made, in order to calculate bayesian surprise, is very limiting: a hypothesis is either true of false (no graded correctness) with respect to data. I think when we do science, we interpret data subjectively, and many times we see data that is inconclusive / confusing with respect to a hypothesis.

---

> ### Author Rebuttal · Authors · 2025-07-31
>
> We sincerely appreciate the time you have taken to review our work! Below, we address each of your concerns and questions.
>
> ---
>
> > `W1a: …results to be very weak in supporting the main message of the paper…I do not see how section 5.1…contribute to the main story…`
>
> Our main contribution is in developing a feasible computational framework to repeatedly sample hypotheses with LLMs that lead to scientific discoveries. Thus, we argue that the choice of which search algorithm drives our framework is indeed a key consideration, besides what reward function guides search (which we evaluate in Section 5.2). Amongst the choices lie various methods that have been shown to be strong candidates in several NLP tasks. For e.g., temperature-based ancestral sampling is often preferred when sampling textual sequences from LLMs despite the existence of more sophisticated techniques such as beam search and MCTS. Therefore, we contend that it remains a non-trivial result that MCTS in our setting of hypothesis generation is the best performer, and, therefore, does contribute meaningfully to our message.
>
> ---
>
> > `W1b: …In section 5.2, only 3 human participants participate in evaluation. No error bars are provided…`
>
> > `Q4: What do the error bars look like for table 1, given that there are only 3 human evaluators? How was inter-annotator agreement calculated? Are the human evaluators the authors themselves? Did the evaluators know which method was used to generate each hypothesis?...`
>
> We clarify that the human study described in section 5.2 is conducted not with 3 humans participants in total, but 3 participants per hypothesis. We provide a longer description of our human study in Section 4 Human Annotations in our supplementary material. The annotators were sourced from the Prolific platform, with a qualification of an MS/PhD STEM degree in either Mathematics and statistics, Information and Communication Technologies, Engineering, manufacturing and construction, or Natural sciences. We also required the residence of annotators to be US, UK, or Canada, and fluency in English. The annotators were not made aware of what method was used to generate the hypotheses.
>
> Thank you for pointing out the lack of error bars in our results. We have now computed 95% confidence intervals using bootstrapping with 10,000 re-samples, resulting in a mean LLM agreement of 0.67 with 95% CI intervals [0.63, 0.71]. We will add these in the final version of the paper.
>
> For inter-annotator agreement, we compute intraclass correlation (ICC) per familiarity class (low/medium/high) using the pingouin python package, as reported in Table 1.
>
> ---
>
> > `W1c: …And given how LLM-heavy the method is, I expect evaluations on more LLMs, not just a single one (gpt-4o).`
>
> We provide analyses with one additional model - o4-mini - in our supplementary material. We show its comparison with GPT-4o in Section 6 “Results with o4-mini (“reasoning” models)” and Figures 5 and 6. In particular, we find that the same trend across search methods holds for both models. With o4-mini, we additionally observe that the gap between MCTS and greedy reduces; quantitatively, we find more number of surprisals than with GPT-4o; qualitatively, we find the hypotheses proposed by o4-mini to involve more complex statistical analyses.
>
> For e.g., the following is a level 5 node found from the Freshwater Fish dataset (DiscoveryBench):
> *“Within South American freshwater-fish sub-basins, evolutionary rates (diversification and morphological evolution) exhibit significant positive spatial autocorrelation, such that geographically proximate basins have more similar rates than distant ones.”*
>
> Due to budget constraints, we are unable to provide analyses with any other models.
>
> ---
>
> > `W2: The assumptions made, in order to calculate bayesian surprise, is very limiting: a hypothesis is either true of false (no graded correctness) with respect to data. I think when we do science, we interpret data subjectively, and many times we see data that is inconclusive / confusing with respect to a hypothesis.`
>
> We clarify that our belief elicitation mechanism does indeed produce graded correctness. Note that while the individual Bernoulli samples from the LLM provide a boolean response, we generate n=30 samples from the LLM. These Bernoulli samples allow us to compute a Beta distribution through the Beta-Bernoulli conjugacy resulting in mean belief probabilities that lie anywhere between 0 and 1. To show that this also happens in practice, here is the histogram (bin count=10; bin width=0.1) of mean prior and mean posterior belief probabilities elicited from the LLM across ~600 hypotheses from runs on 4 datasets:
>
> **Prior Beliefs:**
> Histogram: (array([ 80,  25,  20,  20,  21,  33,  35,  37,  58, 280]), array([0. , 0.1, 0.2, 0.3, 0.4, 0.5, 0.6, 0.7, 0.8, 0.9, 1. ]))
> Mean & std.: 0.6818 ± 0.3396
>
> **Posterior Beliefs (after Bayesian updates):**
> Histogram: (array([  0,   0,  13,  73, 150, 308,  40,  25,   0,   0]), array([0. , 0.1, 0.2, 0.3, 0.4, 0.5, 0.6, 0.7, 0.8, 0.9, 1. ]))
> Mean & std.: 0.4913 ± 0.0929
>
> As shown, our procedure is able to yield graded beliefs, and not 0s and 1s only. Below, we also show the **human prior beliefs** for the same set of hypotheses for comparison:
> Histogram: (array([  1,   0,   5,   7,  28, 102, 129, 176, 115,  46]), array([0. , 0.1, 0.2, 0.3, 0.4, 0.5, 0.6, 0.7, 0.8, 0.9, 1. ]))
> Mean & std.: 0.7067 ± 0.1411
>
> ---
>
> > `Q1: How would the proposed method work with a hypothesis like "the coin is unbiased" and observe coin flips?`
>
> AutoDS is well-suited to handle such a hypothesis. In particular, let’s assume that we have a dataset of coin flips available with a column indicating whether the coin came up heads or tails and each row corresponding to a single coin flip. Now, given the hypothesis “the coin is unbiased”, AutoDS might propose a verification experiment that uses the binomial test (using scipy.stats.binomtest) to check the null hypothesis that the probability of heads is 0.5. This test will then output a p-value, on the basis of which AutoDS will decide whether the hypothesis is supported or not, proceeding to elicit a posterior belief given the hypothesis and evidence from the verification experiment.
>
> ---
>
> > `Q2: Displaying more qualitative results on how the proposed method is doing compared to other baselines / ablated methods would be good for the paper.`
>
> We clarify that our experiments do not reveal significant qualitative variations between different choices of search algorithms (e.g., MCTS, greedy, repeated sampling, etc.) used within AutoDS. This is due to the fact that the same agentic framework is used by all methods as the inner-loop in AutoDS. However, the key difference between different search methods is in the ability to repeatedly find new, surprising hypotheses as described in Figure 2. It is worth noting that where we do see qualitative differences are when the underlying LLM is changed. In Section 6 Results with o4-mini (“reasoning” models) within our supplementary material, we describe how the complexity of statistical analyses increases when using o4-mini, which is a reasoning model from OpenAI, versus using GPT-4o.
>
> ---
>
> > `Q3: Human evaluation is done on the hypotheses generated on only 4 out of 21 datasets. What are these 4 datasets? What do they look like?`
>
> We use SEA-AD, BLADE-Hurricane, BLADE-Boxes, and DiscoveryBench-FreshwaterFish. Our choice was motivated by covering each of the data sources used in our study, while balancing costs for the human study (thus limiting to 4). Here are the natural language descriptions and example columns from each dataset:
>
> **SEA-AD:**
> **Description:** Dataset containing information on donors including demographic details, cognitive status, and neuropathological assessments related to Alzheimer's disease and other conditions.
> **Example columns:** APOE Genotype (a genetic risk factor for Alzheimer's disease), Age at Death, Fresh Brain Weight
>
> **BLADE-Hurricane:**
> **Description:** The dataset is from the Simonsohn et al's “Specification curve analysis” paper published in Nature. It includes archival data on fatalities caused by hurricanes in the United States (1950–2012). Ninety-four Atlantic hurricanes made landfall in the United States during this period.
> **Example columns:** alldeaths (Total number of deaths caused by the hurricane), gender_mf (Binary gender indicator of the hurricane name (0 for male, 1 for female)), wind (Maximum wind speed of the hurricane at the time of landfall in the United States from the NOAA)
>
> **BLADE-Boxes:**
> **Description:** In this study, researchers investigated how children's social learning strategies develop across age in diverse cultural contexts. Social learning, or learning from others, is a key aspect of human cognition and cultural transmission. Children can rely on social information to varying degrees, and they may preferentially learn from majority demonstrations over minority ones. Understanding how these social learning biases emerge and change with age, and how they might be influenced by cultural context, is important for shedding light on the development of human cognition and the emergence of cultural diversity.
> **Example columns:** majority_first (whether the majority option was demonstrated first), gender, age
>
> **DiscoveryBench-FreshwaterFish:**
> **Description:** This dataset contains the drivers of speciation rates in South American freshwater fishes, employing an integrative approach that considers multiple biotic and abiotic factors.
> **Example columns:** soil_div (measures the diversity of soil types or conditions within each sub-basin studied), RML_evol (Rate of Relative Maxillary Length evolution), BEL_evol (Rates of Body Elongation evolution)

---

> > ### Comment · Reviewer_7cwK · 2025-07-31
> >
> > Thank you for the concrete and easy-to-parse responses
> >
> > My w1b/q4, w1c, w2, q1, q2, and q3 are fully addressed. I realized I misunderstood parts of the paper when I mistakenly raised w2 and q1 -- sorry about this.
> >
> > My w1a are partially addressed. I understand that search algorithms are an important detail to getting it to work, but I do not think it should be the first section in your experiment section. I think the paper should first convince the readers that the proposed objective is worth optimizing (as section 5.2 is trying to do) before putting down details on how to optimize it well.
> >
> > In any case, since most of my concerns are addressed, **I'm raising the rating from 3 to 4**. Other things that prevent me from giving even higher scores include the complexity of the domains used (also related to w1a) (they are not simple, and they are a good start, but they are not particularly impressive).

---

> > > ### Author Response · Authors · 2025-08-04
> > >
> > > We're very glad our rebuttal has addressed most of your concerns and appreciate you raising your score.
> > >
> > > ---
> > >
> > > > `...complexity of the domains used (also related to w1a) (they are not simple, and they are a good start, but they are not particularly impressive).`
> > >
> > > We emphasize that the 21 datasets in our study were selected to cover a diversity of domain such as biology, economics, finance, and behavioral science. Importantly, 14 of the 21 datasets were sourced directly from top-tier publications in one of these domains. We submit, therefore, that these fairly represent canonical real-world examples of research datasets that AutoDS is well-suited to be used with. Below, we list the source research article for each of these datasets:
> > >
> > > | Dataset               | Source Paper & Journal                                                                                                      |
> > > |-----------------------|-----------------------------------------------------------------------------------------------------------------------------|
> > > | Freshwater-Fish       | *Accelerated body size evolution in upland environments is correlated with recent speciation in South American freshwater fishes* (Nature, 2023) |
> > > | NLS-BMI, NLS-SES, NLS-Incarceration | National Longitudinal Survey of Youth 1997                                                                    |
> > > | SEA-AD                | *SEA-AD is a multimodal cellular atlas and resource for Alzheimer's disease* (Nature, 2024)                                 |
> > > | Affairs               | *A theory of extramarital affairs* (Journal of Political Economy, 1978)                                                     |
> > > | AMTL                  | *A comparison of antemortem tooth loss in human hunter-gatherers and non-human catarrhines* (American Journal of Physical Anthropology, 2013) |
> > > | Boxes                 | *The development of human social learning across seven societies* (Nature, 2018)                                           |
> > > | Crofoot               | *Interaction location outweighs the competitive advantage of numerical superiority in Cebus capucinus intergroup contests* (PNAS, 2008) |
> > > | Hurricane             | *Female hurricanes are deadlier than male hurricanes* (PNAS, 2014)                                                          |
> > > | Mortgage              | *Mortgage Lending in Boston: Interpreting HMDA Data* (The American Economic Review, 1996)                                  |
> > > | Reading               | *The Impact of Web Browser Reader Views on Reading Speed and User Experience* (CHI, 2019)                                  |
> > > | Soccer                | *Many Analysts, One Data Set: Making Transparent How Variations in Analytic Choices Affect Results* (AMPPS, 2018)          |
> > > | Teaching Ratings      | *Beauty in the classroom: Instructors' pulchritude and putative pedagogical productivity* (Economics of Education Review, 2005) |
> > >
> > > ---
> > > **References:**
> > > Table 3 in *Discoverybench: Towards data-driven discovery with large language models (Majumder et al., 2025)*
> > > Table 3 in *BLADE: Benchmarking Language Model Agents for Data-Driven Science (Gu et al., 2024)*

---

> ### Comment · Reviewer_7cwK · 2025-08-06
>
> Thank you the authors for the reply!
>
> I will take into account these details during the reviewer-AC discussion.
>
> At the same time, I think the domains are not complex in the sense that they are static benchmarks, and they are focused on a specific type of scientific discovery: data-driven discovery. The domains used are not interactive.
>
> The hypotheses generated are also not too complex, such as "Donors who have undergone neuroimaging will have a higher agreement between clinical diagnosis and neuropathological findings for Alzheimer's disease compared to those who have not undergone neuroimaging." Like if the hypothesis is true, it might be a novel finding, but it is not a novel "concept".
>
> (And I still do not think the writing flow of the experiment section is as good as it can be)
>
> That being said, I agree that the domains used in this paper are not simple, and the results present good contribution to the field.

---

> ### Author Response · Authors · 2025-08-06
>
> Thank you for your comments! We appreciate you engaging in an extended discussion with us.
>
> > `At the same time, I think the domains are not complex in the sense that they are static benchmarks, and they are focused on a specific type of scientific discovery: data-driven discovery. The domains used are not interactive.`
>
> We completely agree that moving towards (a) a non-stationary setting where incremental findings affect how later findings are evaluated, and (b) interactive evaluations with feedback, represent exciting directions for the future of this work to move us further towards real-world research workflows. Nonetheless, we found making these simplifying assumptions necessary to make initial progress towards our longer-term goal.
>
> ---
>
> > `...The hypotheses generated are also not too complex...`
>
> Thank you for raising this concern and citing an example from Appendix A1. It is true that *surprisals*, as accumulated by AutoDS, may or may not turn out to be actual discoveries. The cited example, for instance, may well turn out to be a known concept for a domain expert. However, our main hypothesis is that by optimizing for Bayesian surprise and collecting several such findings, we are more likely to find instances that may turn out to be real discoveries, versus optimizing for other automatic metrics (which we have taken a first step at validating with a human study in Table 1).
>
> Regarding complexity, we find that this is often a function of the underlying LLM being used. As the underlying LLMs get more powerful, the complexity of the generated hypotheses increases. For example, below are a few hypotheses that were found surprising from the SEA-AD dataset (same as the cited example) when using o4-mini instead of GPT-4o:
>
> *"Higher educational attainment attenuates the positive association between composite neuropathological burden and the odds of dementia in older adults, after adjusting for age at death and sex."*
>
> *"Higher neuropathological burden (Thal phase, Braak stage, CERAD score) and APOE ε4 carrier status are associated with an increased hazard (earlier onset) of cognitive symptoms, with sex also significantly predicting hazard in a Cox proportional hazards framework."*
>
> *"In a 1:1 propensity-matched cohort of Alzheimer's disease and control donors balanced on Age at Death, Sex, PMI, and RIN, donors with Alzheimer's disease will exhibit lower Brain pH compared to matched controls."*
>
> *"As the categorical severity of Alzheimer's disease neuropathology increases from Not AD to High, the normalized brain weight index (fresh brain weight divided by age at death) will decrease."*
>
> We see a marked increase in complexity (also seen in the proposed experiment plans for these hypotheses; omitted here for brevity but we can provide them in a follow-up message, if you'd like).
>
> ---
>
> > `...I still do not think the writing flow of the experiment section is as good as it can be...`
>
> Thank you for bringing up this criticism again. Our response earlier missed addressing it! We agree that swapping the order in which we describe the results will benefit exposition, i.e., in the final version of the paper, we will first discuss the efficacy of Bayesian surprise determined via our human study and then compare the performance of the search algorithms. Thank you for your suggestion!
>
> ---
>
> We hope our discussion is helpful as you make your final decision and truly appreciate your time!

---

> > ### Comment · Reviewer_7cwK · 2025-08-06
> >
> > Thank you the authors for all concrete replies. I agree this paper is a good first step, setting things up for future work, such as incremental hypothesis building (which should output hypotheses that are more complex than direct LLM prompting), interactive evaluation, etc. And thank you for addressing my concern of the flow of the experiment section. I maintain my raised, positive score of 4, but with higher confidence!

---

> > > ### Author Response · Authors · 2025-08-06
> > >
> > > We appreciate your time and engagement. Thank you again!

---

### Official Review · Reviewer_8GRz · 2025-07-05

**Clarity:** 2
**Significance:** 3
**Originality:** 2
**Rating:** 4
**Confidence:** 3

**Summary:**

This paper introduces AUTODS, a novel framework for open-ended autonomous scientific discovery (ASD) that leverages Bayesian surprise as a guiding principle to explore hypotheses without predefined goals. While prior work in ASD often requires human-specified questions, AUTODS allows a language model to autonomously propose, test, and evaluate hypotheses based solely on data, thereby simulating the behavior of an independent scientific agent.The core idea is to quantify epistemic shifts—changes in the model’s belief before and after observing experimental results—using Bayesian surprise. The method formalizes belief elicitation via sampling, applies a Beta-Bernoulli model for prior/posterior estimation, and defines a surprisal score to capture meaningful belief updates. To efficiently explore the hypothesis space, AUTODS integrates Monte Carlo Tree Search (MCTS) with progressive widening, using surprisal as the reward signal to balance exploration and exploitation.Experiments across 21 real-world datasets from domains including biology, economics, and behavioral science show that AUTODS outperforms strong baselines—producing 5–29% more discoveries judged surprising by the LLM. A human study found that 67% of these LLM-surprising discoveries also surprised human experts, confirming the alignment between Bayesian surprise and human intuition. The authors also compare Bayesian surprise with other automatic reward metrics (e.g., LLM interestingness/utility) and validate AUTODS’s internal experiment execution and deduplication mechanisms.In conclusion, the paper presents a promising and scalable approach to fully autonomous, open-ended scientific discovery by marrying principled Bayesian reasoning with powerful language models.

**Questions:**

1. How does the system respond to misleading or incorrect data? Is it susceptible to being misled by "spurious surprise"?

2. How can the model's ability to transition from "discovering correlations" to "inferring causality" be improved?

3. Can this method be extended to literature-driven open discovery, rather than being limited to data-driven discovery?

**Ethical Concerns:**

["NO or VERY MINOR ethics concerns only"]

**Final Justification:**

Primary concerns have been resolved

**Limitations:**

Despite proposing an innovative approach in the field of open-ended scientific discovery, this paper still has some limitations. First, its "surprise" metric relies on the belief changes of the language model itself, which is subjective and may not always represent truly valuable scientific discoveries. Second, due to the limited context window of the language model, its reasoning ability is restricted as the exploration deepens. In addition, the current verification process mainly depends on simple statistical analysis and is not yet capable of handling complex scientific mechanisms or causal reasoning problems. The method is computationally expensive, with each hypothesis node taking a long time on average, which limits the efficiency for large-scale applications. Finally, the system is sensitive to data quality and tends to discover "surprising" positive conclusions, potentially overlooking negative findings or the closed loop of scientific validation.

**Paper Formatting Concerns:**

This paper has no formatting concerns.

**Quality:**

3

**Strengths And Weaknesses:**

**Strengths**

1. **Well-Designed Methodology**: Effectively balances exploration and exploitation by integrating Monte Carlo Tree Search (MCTS) with Bayesian surprise.

2. **Outstanding Empirical Results**: Significantly outperforms existing methods on 21 real-world datasets.

**Weaknesses / Limitations**

1. **Dependent on Subjective Beliefs Expressed by LLMs**: The model's "surprise" does not equate to objective "novelty" or "importance," and may be biased.

2. **Limited Depth of Reasoning**: Currently focuses on shallow statistical hypotheses and has not demonstrated the ability to model complex scientific mechanisms.

3. **High Computational Overhead**: Each node requires an average of 75 seconds for processing, resulting in a relatively slow overall search speed.

---

> ### Author Rebuttal · Authors · 2025-07-31
>
> Thank you for the time you have taken to review our work! Please see our responses below to each of your comments and questions.
>
> ---
>
> > `W1: The model's "surprise" does not equate to objective "novelty" or "importance"`
>
> Indeed, “surprising” hypotheses need not always be novel or important. We, therefore, conducted an extensive human study to assess how many surprisals from our system are indeed “interesting” and of “high utility”. We find that, when optimized for Bayesian Surprise, 73% of hypotheses deemed surprising were also attributed as “interesting” and 79% of the surprising hypotheses were of “high utility” according to STEM MS/PhDs. We detail these experimental results in Section 5.2 from Line 284 and Table 2.
>
> ---
>
> > `W2: Shallow statistical hypotheses`
>
> We find that the richness of statistical hypotheses is a function of the underlying LLM. For example, as described in Section 6 of our supplementary material (L176), we find, qualitatively, that the complexity of hypotheses generated by o4-mini is significantly higher than GPT-4o.
>
> E.g., the following is a level 5 node found from the Freshwater Fish dataset (DiscoveryBench):
> *“Within South American freshwater-fish sub-basins, evolutionary rates (diversification and morphological evolution) exhibit significant positive spatial autocorrelation, such that geographically proximate basins have more similar rates than distant ones.”*
>
> This hypothesis was verified by the following experimental plan:
> 1. Compute global and local spatial autocorrelation (Moran's I, LISA) for the key response variables: diversification rate (DR, BAMM_NetDiv) and a morphological composite (e.g., the PC1 from PCA on BEL_evol–RML_evol).
> 2. Construct a spatial weights matrix (e.g., k‐nearest neighbors or distance‐based) from the lat/long coordinates using PySAL.
> 3. Report global Moran's I and its significance for each variable, then map local Moran's I clusters to identify hotspots and coldspots of high/low evolutionary activity across basins.
>
> ---
>
> > `W3: High computational overhead`
>
> In section 5.4, while we highlight that the average processing time per node (hypothesis generation to verification) of 75 seconds may be improved by an initial programmatic search, we emphasize that AutoDS, nonetheless, provides a massive improvement over the average time a human researcher takes to perform all steps in a node. Gu et al. (2024) highlight that data analysts often take ~20 minutes to even detail their analysis plan before implementing standard data analysis tasks. Further, Majumder et al. (2025) show that it can take 90 human hours to reasonably replicate complex data-driven scientific studies from documentation and open-sourced code. Moreover, we expect that the gap between the time taken by a human vs. AutoDS to generate and verify a new hypothesis and experiment increases with complexity. We argue, therefore, that 75 seconds per node represents very high throughput for autonomously generating and verifying scientific hypotheses in data-driven discovery compared to a completely manual undertaking.
>
> **References:**
> *How Do Data Analysts Respond to AI Assistance? A Wizard-of-Oz Study. Gu et al., 2024.
> DiscoveryBench: Towards Data-Driven Discovery with Large Language Models. Majumder et al., 2024.*
>
> ---
>
> > `Q1: How does the system respond to misleading or incorrect data? Is it susceptible to being misled by "spurious surprise"?`
>
> We argue that any research undertaking (human or machine) is only as good as the quality of the data sources used in the research. AutoDS is not an exception to this—if the input dataset is noisy, each of hypothesis generation, verification, and analysis will accumulate errors, leading to failure. Regarding spurious surprise, we find that priors encoded within the LLM from pre-training as well as instruction-following skills imbued at post-training lend significant robustness to this error mode. In our experiments, we do not encounter instances where the model attempts to “reward hack” to generate contrived hypotheses that would always lead to surprisal.
>
> ---
>
> > `Q2: How can the model's ability to transition from "discovering correlations" to "inferring causality" be improved?`
>
> This is a great question! We believe there are several ways this transition may be made. First, if the input dataset itself is large enough, predictive models to verify hypotheses may be trained on a train subset of the dataset and then further evaluated on an unseen test split. Second, verification may be conducted on additional datasets (our implementation already supports taking in multiple data sources). Third, equipping the model with knowledge of causal modelling via existing programmatic constructs may be done explicitly. For e.g., this may be provided as additional instructions to the programmer agent in the inner-loop agentic framework.
>
> ---
>
> > `Q3: Can this method be extended to literature-driven open discovery, rather than being limited to data-driven discovery?`
>
> Indeed! Our framework provides natural extensions to incorporating literature in two places. During hypothesis generation, relevant papers to the node selected by MCTS may be retrieved to inform subsequent generation. During belief elicitation, papers relevant to the hypothesis being evaluated can be retrieved as additional evidence and injected in-context before generating Bernoulli belief samples.
>
> ---
>
> > `Limitations: …tends to discover "surprising" positive conclusions, potentially overlooking negative findings or the closed loop of scientific validation.`
>
> We disagree with this conclusion. Our experiments show that AutoDS is able to surface positive and negative findings in equal measure. Please see the following generations as examples of negative findings that AutoDS discovered:
> *“There is no significant difference between the property damage estimates adjusted to 2013 and 2015 monetary values”*
> *“There is no significant difference in the total number of deaths between hurricanes with male and female names.”*
> *“There is no significant correlation between the last MOCA score and brain pH levels in the dataset.”*

---

> > ### Comment · Reviewer_8GRz · 2025-08-06
> > **Response to rebuttal**
> >
> > Thank you for the clarification, which has addressed most of my concerns.
> >
> > I maintain my positive evaluation of this work.
> >
> > Given the paper's focus on scientific discovery, I recommend including discussion of prior work on early hypothesis generation—for example, "Large language models as zero-shot hypothesis proposers" would be a relevant reference to incorporate.

---

> > > ### Author Response · Authors · 2025-08-06
> > >
> > > We appreciate your vote of confidence. Thank you also for the citation! We will surely include it in our discussion in Related Works.

---

### Official Review · Reviewer_wYSZ · 2025-07-06

**Clarity:** 3
**Significance:** 3
**Originality:** 3
**Rating:** 5
**Confidence:** 4

**Summary:**

This paper addresses the critical challenge of open-ended autonomous scientific discovery (ASD), where an AI system must decide which hypotheses to investigate without human guidance. The authors propose AUTODS, a novel framework that moves beyond ill-defined heuristics like "interestingness" and instead uses the principled metric of Bayesian surprise to guide exploration. Bayesian surprise is formalized as the KL-divergence between an LLM's prior and posterior beliefs about a hypothesis, before and after observing experimental evidence from an auto-generated verification procedure. To efficiently navigate the vast hypothesis space, AUTODS employs a Monte Carlo Tree Search (MCTS) algorithm that uses this surprisal metric as its reward function. The method is rigorously evaluated on 21 real-world datasets, demonstrating superior performance in discovering surprising findings compared to strong search baselines. Crucially, a human study validates that the model's "surprise" correlates strongly with that of human experts, marking a significant step towards truly autonomous scientific exploration.

**Questions:**

The framework equates surprising the LLM with scientific discovery. How does the system defend against simply rediscovering facts that are known to human experts but were absent in the model's training data? Could the surprisal metric be augmented with a "novelty to humanity" check (e.g., via targeted literature or web search) to better align the model's discovery frontier with the human scientific frontier?

The Beta-Bernoulli model for belief elicitation is a key component. How sensitive are the resulting surprisal scores and the MCTS search performance to the choice of both the elicitation model (e.g., GPT-3.5 vs. GPT-4o) and the number of samples (n) used for estimation? Please provide analysis on the stability and variance under these differing conditions.

MCTS is well-motivated, but could the surprisal reward incentivize the agent to find "gotchas" or quirky statistical outliers rather than pursuing a coherent and systematically surprising research direction? How does the system ensure a high surprisal on a single node translates to a fruitful research branch, rather than a dead end?

For reproducibility, could you please report the key MCTS hyperparameters used in your experiments (specifically, the exploration constant C, and the progressive widening parameters k and alpha) and provide brief justification for their selection?

**Ethical Concerns:**

["NO or VERY MINOR ethics concerns only"]

**Final Justification:**

I re-read the paper and rebuttal and keep my score at 5 (Accept), confidence 4. The authors addressed my main questions: (i) the “binary” belief concern—belief elicitation is graded via Beta-Bernoulli with n=30 (GPT-4o) / n=8 (o4-mini) at T=0.7, and their sensitivity analysis shows low variance in surprisal rates as n varies; (ii) reproducibility—MCTS hyperparameters are now reported (C=1, progressive-widening k=1, α=0.5) with rationale; (iii) robustness—results with a second model (o4-mini) preserve the trends and yield more complex hypotheses; and (iv) reward hacking—the surprisal signal sits in the outer loop while hypothesis generation remains decoupled, which mitigates “gotcha” optimization in practice. The human study is also strengthened with a 95% CI for LLM–human surprisal agreement (~0.67, CI [0.63, 0.71]).

I still view two limitations as worth highlighting for the camera-ready: (1) Compute/latency (~75s per node) and heavy multi-call dependence, even if the throughput compares favorably to manual workflows; and (2) the “observer” problem—surprising the LLM is not identical to novelty-to-humanity. The paper already sketches literature-grounded extensions; I encourage adding a lightweight “novelty-to-human” check (targeted retrieval or citation search) and a brief note on dataset-selection/open-endedness. A larger multi-model sweep and a broader human study would further solidify the claims but are not blockers. Overall, the work is well-motivated, carefully engineered, and empirically convincing; I recommend accept.

**Limitations:**

The authors provide a good error analysis in the appendix and discuss the latency of their approach. For a more complete discussion, they should also explicitly address the risks of LLM hallucinations in the code generation and analysis steps, and the broader ethical implications of deploying autonomous agents that could generate and promote potentially spurious scientific hypotheses without sufficient human oversight.

**Paper Formatting Concerns:**

None.

**Quality:**

3

**Strengths And Weaknesses:**

Strengths

1. Principled Guiding Metric
The paper's primary strength is its conceptual shift from ambiguous proxies like "interestingness" to the formally defined and theoretically grounded metric of Bayesian surprise. This provides a robust and replicable foundation for guiding open-ended discovery.

2. Novel and Effective Synthesis
The authors impressively synthesize three complex components into a coherent system (1) a practical method for eliciting Bayesian belief distributions from LLMs, (2) the use of KL-divergence as a reward signal, and (3) a sophisticated MCTS with progressive widening to effectively balance exploration and exploitation. This combination is both novel and demonstrably effective.

3. Exceptional Empirical Validation
The evaluation is comprehensive and convincing. The system's outperformance of strong baselines across 21 diverse datasets is significant. The inclusion of a human evaluation study is a critical and standout feature, providing strong evidence that optimizing for the LLM's surprisal is a valid proxy for discovering findings that are surprising to human domain experts. The careful validation of the agent's internal components, such as experiment correctness and deduplication, further strengthens the work's quality.

Weaknesses

1. The "Observer" Problem
The framework's goal is defined as expanding the LLM's own knowledge frontier. This pragmatically conflates the LLM's surprise—an artifact of its training data's scope—with genuine scientific surprise. The system may therefore be surprised by a fact that is well-established within a niche scientific field but absent from its general training corpus, potentially leading it to "rediscover" known science.

2. Fidelity and Robustness of Belief Elicitation
The belief elicitation process models the LLM's complex epistemic state with a Beta distribution estimated from a small number of boolean samples from a single proprietary model (GPT-4o). This elegant simplification may not fully capture the nuances of a model's confidence, and its robustness to smaller, open, or domain-specific models is untested.

3. Computational Intensity
The search process is inherently computationally expensive, with each node in the MCTS tree requiring multiple LLM calls for hypothesis generation, verification, and belief elicitation (reported as 75 seconds/node on average). While the method is more efficient than baselines, this high absolute cost may limit its scalability for truly massive discovery campaigns.

---

> ### Author Rebuttal · Authors · 2025-07-31
>
> Thank you for the time you have taken to review our work! Below, we address each of your concerns and questions.
>
> ---
>
> > `W1: The "Observer" Problem…to "rediscover" known science.`
>
> > `Q1: Could the surprisal metric be augmented…via targeted literature or web search…with the human scientific frontier?`
>
> Indeed, the performance of AutoDS depends on the quality of priors that are encoded in the LLM with respect to the domain of interest. However, the main contribution of our work is in proposing a computational framework that is useful for scientific discovery. Our vision is that practitioners will use AutoDS with custom models—pre-trained or fine-tuned on their domains of interest—to align discovery with their research goals. Furthermore, AutoDS is not only applicable to the data-driven discovery setting, but has a natural extension to literature-driven discovery (as also suggested in your question). This offers another avenue to prevent the re-discovery problem by grounding both hypothesis generation and belief elicitation in a retrieved set of relevant papers, which would provide additional domain knowledge that goes beyond model pre-training.
>
> ---
>
> > `W2: Fidelity and Robustness of Belief Elicitation…may not fully capture the nuances of a model's confidence`
>
> Despite the simplicity of the Bernoulli-Beta belief formulation we use in AutoDS, we do find that it is sufficiently robust in providing alignment with human beliefs. As described in Section 5.2 (and Section 4 in the supplementary material), we test this by conducting a human study across 4 datasets. Our analysis finds that human surprisal has a correlation with LLM surprisals found via the Beta-Bernoulli belief elicitation in 67% of the hypotheses. Additionally, the 95% confidence interval is [0.63, 0.71], computed using bootstrapping with 10,000 re-samples (we will include this detail in the final version of the paper).
>
> ---
>
> > `Q2: How sensitive are the resulting surprisal scores and the MCTS search performance to the choice of both the elicitation model (e.g., GPT-3.5 vs. GPT-4o) and the number of samples (n) used for estimation?...`
>
> We have included a comparison between GPT-4o and o4-mini in our supplementary material (Section 6 “Results with o4-mini (“reasoning” models)” and Figures 5 and 6). In particular, we find that the same trend across search methods holds for both models. With o4-mini, we additionally observe that the gap between MCTS and greedy reduces; quantitatively, we find more number of surprisals than with GPT-4o; qualitatively, we find the hypotheses proposed by o4-mini to involve more complex statistical analyses.
>
> On your suggestion, we have now run a sensitivity analysis of the number of samples used for belief elicitation on the downstream cumulative surprisal count. To do this analysis, we sample 100 hypotheses at random from the MCTS run on the Seattle-Alzheimer’s (SEA-AD) dataset. We then regenerate prior and posterior beliefs for these hypotheses using different numbers of samples and report the proportion of hypotheses found surprising in each case. Our results are as follows:
>
> | Samples | Surprisal Proportion |
> |---------|----------------------|
> | 1       | 0.9531                 |
> | 5       | 0.9219                 |
> | 10      | 0.8906                 |
> | 20      | 0.9219                 |
> | 30      | 0.8906                 |
> | 40      | 0.8906                 |
> | **Aggregate** | **0.9115 ± 0.0233 (std.)**         |
>
> As indicated by the low standard deviation, we see only modest variation in the final surprisal proportion across settings of the number of samples used for belief elicitation.
>
> ---
>
> > `W3: …While the method is more efficient than baselines, this high absolute cost may limit its scalability for truly massive discovery campaigns.`
>
> In section 5.4, while we highlight that the average processing time per node (hypothesis generation to verification) of 75 seconds may be improved by an initial programmatic search, we emphasize that AutoDS, nonetheless, provides a massive improvement over the average time a human researcher takes to perform all steps in a node. Gu et al. (2024) highlight that data analysts often take ~20 minutes to even detail their analysis plan before implementing standard data analysis tasks. Further, Majumder et al. (2025) show that it can take 90 human hours to reasonably replicate complex data-driven scientific studies from documentation and open-sourced code. Moreover, we expect that the gap between the time taken by a human vs. AutoDS to generate and verify a new hypothesis and experiment increases with complexity. We argue, therefore, that 75 seconds per node represents very high throughput for autonomously generating and verifying scientific hypotheses in data-driven discovery compared to a completely manual undertaking, and does not, therefore, present a barrier to larger discovery campaigns. We believe that the barrier comes from being able to consistently generate useful hypotheses, which we aim to address in our paper.
>
> **References:**
> *How Do Data Analysts Respond to AI Assistance? A Wizard-of-Oz Study. Gu et al., 2024.
> DiscoveryBench: Towards Data-Driven Discovery with Large Language Models. Majumder et al., 2024.*
>
> ---
>
> > `Q3: …could the surprisal reward incentivize the agent to find "gotchas" or quirky statistical outliers... How does the system ensure a high surprisal on a single node translates to a fruitful research branch, rather than a dead end?`
>
> Reward hacking is indeed important to guard against in a system like AutoDS. We argue that the modular agent structure in our system provides a natural safeguard against this behavior. That is, in AutoDS, the MCTS search mechanism, which utilizes surprisal-based rewards, is what is used to prioritize which nodes amongst previously generated ones should be selected for further expansion (new hypothesis generation). However, the hypothesis generation agent, itself, operates independent of the knowledge of this surprisal reward. The diversity in new hypotheses thus comes from conditioning on diverse branches of the tree, and not a direct signal to optimize for the surprisal reward, which is only used by the outer-loop.
>
> It is worth mentioning, additionally, that AutoDS, too, cannot ensure that a node translates to a fruitful research branch. However, using the hypothesis that nodes that are surprising are likely to lead to further surprisals, we use MCTS to prioritize surprisal nodes for expansion. As shown in our main results, this indeed leads to the most number of cumulative surprises across 21 datasets vs. using random sampling or a greedy strategy.
>
> ---
>
> > `Q4: For reproducibility, could you please report the key MCTS hyperparameters used in your experiments (specifically, the exploration constant C, and the progressive widening parameters k and alpha) and provide brief justification for their selection?`
>
> We set exploration constant C=1, as a default setting across datasets to have exploration and exploitation contribute equally. Note that given either prior knowledge about the domain or additional budget to test different settings, C may be tuned to obtain better search performance. For progressive widening, we set 𝛼 = 0.5 and k = 1 as the constant setting across domains. Again, we do not tune this, but could if the budget allows. Our selection aims to get trees of balanced depth and width. We will include these details in the main paper. Thank you!
>
> ---
>
> > `Limitations: …explicitly address the risks of LLM hallucinations in the code generation and analysis steps, and the broader ethical implications of deploying autonomous agents that could generate and promote potentially spurious scientific hypotheses without sufficient human oversight.`
>
> We are committed to responsibly discussing our results and its implications. We address a few of these concerns in Appendix B Error Analysis as well as take a cautionary tone in our conclusion. We will additionally provide an expanded discussion on the risks of automated discovery in our final version of the paper.

---

### Official Review · Reviewer_WY1X · 2025-07-06

**Clarity:** 4
**Significance:** 4
**Originality:** 4
**Rating:** 4
**Confidence:** 4

**Summary:**

This paper introduces AutoDS, a Bayesian framework that emulates the process of scientific discovery by modeling posterior surprisal through Monte Carlo Tree Search. The methodology provides a formal definition of surprise in the context of automated scientific discovery (ASD), characterizing it as the Kullback–Leibler divergence (DKL) between prior and posterior beliefs. To further capture the directional and informational aspects of surprise, the concept is refined as Bayesian surprise under belief shift. Experiments on 21 real-world datasets demonstrate that AutoDS achieves 5–29% more discoveries deemed surprising by a large language model (LLM), outperforming strong search-based baselines.

**Questions:**

1. Is any scientific discovery process truly binary? The assumption that the verification procedure must yield a binary outcome—i.e., supported vs. unsupported—feels overly restrictive. In NLP, for instance [2], researchers have explored the use of p-values reported in papers as a more continuous measure of how strongly evidence supports a hypothesis.

2. What happens if an LLM produces different Boolean responses when given the same prompt multiple times? What temperature did you use when calling the APIs? Would such inconsistency undermine the reliability of the empirical frequency estimates?

3. What $n$ was used for belief elicitation?

[2] Wadden, D., Lin, S., Lo, K., Wang, L. L., van Zuylen, M., Cohan, A., & Hajishirzi, H. (2020, November). Fact or Fiction: Verifying Scientific Claims. In Proceedings of the 2020 Conference on Empirical Methods in Natural Language Processing (EMNLP) (pp. 7534-7550).

**Ethical Concerns:**

["NO or VERY MINOR ethics concerns only"]

**Final Justification:**

While some of my concerns were asked by other reviewers and addressed by the authors, I remain my original scores, as I believe my rating was fair.

**Limitations:**

It would be good to include results from LLMs other than GPT-4o.

**Quality:**

4

**Strengths And Weaknesses:**

I believe the authors address a highly relevant and timely question—shifting the focus from merely discovering novel or diverse scientific hypotheses to leveraging the emergent capabilities of LLMs to guide their own discovery processes. Given the rapid advancement of foundation models, I anticipate that this approach will gain broader traction within the AI4Science community. The paper is clearly written, and the experiments are thoughtfully and comprehensively designed.

One area for improvement would be a more explicit comparison with alternative formulations of uncertainty. For instance, in addition to using the Kullback–Leibler divergence (DKL) between prior and posterior distributions, it would be interesting to consider mutual information as a measure—evaluating how much information is gained after observing the data. (Although these formulations may be mathematically related, their interpretations can differ depending on the application.) I would point to Gao et al. [1] as an example, where mutual information is used to align conclusions with their most probable premises in scientific texts.

That said, the authors provide an in-depth treatment of DKL, which is sufficient for supporting the central idea and validating the proposed methodology.

[1] Gao, Y., Gu, N., Lam, J., Henderson, J., & Hahnloser, R. (2024, March). Evaluating Unsupervised Argument Aligners via Generation of Conclusions of Structured Scientific Abstracts. In Proceedings of the 18th Conference of the European Chapter of the Association for Computational Linguistics (Volume 2: Short Papers), pp. 151–160.

---

> ### Author Rebuttal · Authors · 2025-07-31
>
> Thank you for taking the time to review our work! We address each of your comments and questions below.
>
> ---
>
> > `W1: …interesting to consider mutual information as a measure—evaluating how much information is gained after observing the data…Gao et al. [1]...That said, the authors provide an in-depth treatment of DKL, which is sufficient for supporting the central idea and validating the proposed methodology.`
>
> Thank you for this suggestion and the reference! Mutual information is indeed a related information theoretic concept to surprisal. As you note, there is a mathematical equivalence as well. We can surely add this as a discussion in our related works in the final version of the paper.
>
> ---
>
> > `Q1: Is any scientific discovery process truly binary?...researchers have explored the use of p-values reported in papers as a more continuous measure…`
>
> > `Q2: …What temperature did you use when calling the APIs?...`
>
> We clarify that our belief elicitation mechanism does indeed produce graded correctness. Note that while the individual Bernoulli samples from the LLM provide a boolean response, we generate n=30 samples from the LLM using a temperature of 0.7. These Bernoulli samples allow us to compute a Beta distribution through the Beta-Bernoulli conjugacy resulting in mean belief probabilities that lie anywhere between 0 and 1. To show that this also happens in practice, here is the histogram (bin count=10; bin width=0.1) of mean prior and mean posterior belief probabilities elicited from the LLM across ~600 hypotheses from runs on 4 datasets:
>
> **Prior Beliefs:**
> Histogram: (array([ 80,  25,  20,  20,  21,  33,  35,  37,  58, 280]), array([0. , 0.1, 0.2, 0.3, 0.4, 0.5, 0.6, 0.7, 0.8, 0.9, 1. ]))
> Mean & std.: 0.6818 ± 0.3396
>
> **Posterior Beliefs (after Bayesian updates):**
> Histogram: (array([  0,   0,  13,  73, 150, 308,  40,  25,   0,   0]), array([0. , 0.1, 0.2, 0.3, 0.4, 0.5, 0.6, 0.7, 0.8, 0.9, 1. ]))
> Mean & std.: 0.4913 ± 0.0929
>
> As shown above, our procedure is indeed able to yield graded beliefs - not 0s and 1s only. Below, we also show the **human prior beliefs** for the same set of hypotheses for comparison:
> Histogram: (array([  1,   0,   5,   7,  28, 102, 129, 176, 115,  46]), array([0. , 0.1, 0.2, 0.3, 0.4, 0.5, 0.6, 0.7, 0.8, 0.9, 1. ]))
> Mean & std.: 0.7067 ± 0.1411
>
> Furthermore, our hypothesis verification procedures do produce p-values. However, instead of hard-coding the significance value that should be used to reject the null hypothesis, we instead sample LLM posterior beliefs given the evidence from the result of the statistical tests (including the p-values) to determine the change in belief for the hypothesis.
>
> ---
>
> > `Q3: What n was used for belief elicitation?`
>
> We use 30 for GPT4o, and 8 for o4-mini (we had to reduce the samples for o4-mini because 8 is the maximum number of responses we can generate in a single call). Note that the number of samples can be scaled up to any arbitrary number, though we do not see enough variance when we increase it from our current number for these models. Please see below a sensitivity analysis of the choice of number of samples on the proportion of surprisals found conducted using 100 randomly sampled hypotheses generated from the SEA-AD dataset:
>
> | Samples | Surprisal Proportion |
> |---------|----------------------|
> | 1       | 0.9531                 |
> | 5       | 0.9219                 |
> | 10      | 0.8906                 |
> | 20      | 0.9219                 |
> | 30      | 0.8906                 |
> | 40      | 0.8906                 |
> | **Aggregate** | **0.9115 ± 0.0233 (std.)**         |
>
> ---
>
> > `Limitations: It would be good to include results from LLMs other than GPT-4o.`
>
> We do provide additional results comparing all baselines with another model - o4-mini (a “reasoning” model) - in Section 6 of our supplementary material. We see modest but steady gains in terms of cumulative surprisal counts with reasoning models (o4-mini vs. GPT-4o). Furthermore, qualitatively, we find that the complexity of hypotheses generated by o4-mini is higher than GPT-4o.

---

### Note · Authors · 2025-08-13

Dear reviewers and AC,

We deeply appreciate your service to the community. The constructive feedback, suggestions, and discussion during the review period have been very useful in improving our submission. We re-iterate our commitment to incorporate all writing suggestions, additional related works, discussion on risks, and new analyses in the final version of our manuscript.

Thank you for your thoughtful engagement!

---

### Decision · Program_Chairs · 2025-09-17

**Decision:**

Accept (poster)

**Comment:**

The paper proposes an approach for autonomous scientific discovery using the concept of Bayesian surprise in LLMs. Bayesian surprise is defined as the shift from prior beliefs  about a hypothesis to posterior beliefs after observing data, and is quantified using as the KL divergence between the LLM’s prior and posterior. This  is used as the reward function to drive MCTS. Empirical results show significant gains compared to alternatives under fixed budget conditions. Evaluation confirming 2/3 alignment with human judgements further supports the credibility of the proposed approach, specifically 73% of the hypotheses deemed surprising were also attributed as interesting and 79% were judged to be high quality.

Given the ever increasing capabilities of LLMs, autonomous scientific discovery is highly  relevant and timely as noted by the reviewers. The proposed approach is well motivated,  clearly explained with sufficient  detail. The question about using “surprise” as a proxy to “scientific novelty” is addressed to some extent by the human study to assess how many surprisals suggested by the system are “interesting” or “high quality”. Authors’ rebuttal and discussion that followed helped clarify some points, especially regarding the belief elicitation mechanism about which multiple reviewers were confused. It would be good to incorporate some of the discussion points that helped clarify the approach in the camera ready version.